# Non-shared coding of observed and executed actions prevails in macaque ventral premotor mirror neurons

Jörn K Pomper[1,2][*][†], Mohammad Shams[1,3,4][†], Shengjun Wen[1,3][†], Friedemann Bunjes[1], Peter Thier[1]

[1]Cognitive Neurology Laboratory, Hertie Institute for Clinical Brain Research, University of Tübingen, Tübingen, Germany; [2]Department of Neurology & Stroke, Hertie Institute for Clinical Brain Research, University of Tübingen, Tübingen, Germany; [3]Graduate Training Centre of Neuroscience, International Max Planck Research School, University of Tübingen, 72076 , Germany, Tübingen, Germany; [4]Department of Psychology, York University, Toronto, Canada

**\*For correspondence:**
joern.pomper@uni-tuebingen.de

[†]These authors contributed equally to this work

**Competing interest:** The authors declare that no competing interests exist.

**Abstract** According to the mirror mechanism the discharge of F5 mirror neurons of a monkey observing another individual performing an action is a motor representation of the observed action that may serve to understand or learn from the action. This hypothesis, if strictly interpreted, requires mirror neurons to exhibit an action tuning that is shared between action observation and execution. Due to insufficient data it remains contentious if this requirement is met. To fill in the gaps, we conducted an experiment in which identical objects had to be manipulated in three different ways in order to serve distinct action goals. Using three methods, including cross-task classification, we found that at most time points F5 mirror neurons did not encode observed actions with the same code underlying action execution. However, in about 20% of neurons there were time periods with a shared code. These time periods formed a distinct cluster and cannot be considered a product of chance. Population classification yielded non-shared coding for observed actions in the whole population, which was at times optimal and consistently better than shared coding in differentially selected subpopulations. These results support the hypothesis of a representation of observed actions based on a strictly defined mirror mechanism only for small subsets of neurons and only under the assumption of time-resolved readout. Considering alternative concepts and recent findings, we propose that during observation mirror neurons represent the process of a goal pursuit from the observer's viewpoint. Whether the observer's goal pursuit, in which the other's action goal becomes the observer's action goal, or the other's goal pursuit is represented remains to be clarified. In any case, it may allow the observer to use expectations associated with a goal pursuit to directly intervene in or learn from another's action.

## Editor's evaluation

The mechanisms underlying mirror neurons are a topic of wide interest for all who study the workings of the brain. The authors use an elegant and compelling decoding approach to test whether mirror neurons encode action categories in the same framework underlying action execution regardless of whether actions are executed in the dark or observed in the light. This new approach identifies only a very small subset of mirror neurons with fully matched coding among another set with partial matches and a population-wide code more consistent with representing goal pursuit. The thought-provoking and important study opens up new avenues to probe the neural mechanisms of matching action and perception.

## Introduction

Mirror neurons (MNs) were discovered in 1992 in ventral premotor cortical area F5 of macaque monkeys (*di Pellegrino et al., 1992*). They received their name in 1996 when they were introduced by Gallese and colleagues as neurons that " (...) discharged both when the monkey made active movements and when it observed specific meaningful actions performed by the experimenter." (*Gallese et al., 1996*, p. 595). The peculiarity of this finding was that premotor neurons, whose function had until then been exclusively seen in action control, modulated their discharge rate even when the observer did not have to act at all. In addition, many of these neurons seemed to prefer the same action during observation and execution. This eventually led to the interpretation that 'motor' neurons were found that might be important for imitating (*Rizzolatti et al., 2001*) or understanding an observed action (*Gallese et al., 1996*; *Rizzolatti et al., 1996*). In subsequent years, neurons with similar properties were found in other brain regions and in other species, including humans (*Fogassi et al., 2005*; *Mukamel et al., 2010*). More recently, it has been proposed that there are specific mirror neuron networks for actions, emotions, and vitality, each involving multiple brain regions (*Rizzolatti and Sinigaglia, 2016*). Their commonality would be that they serve to understand others based on a mirror mechanism that has been proposed as a basic principle of brain function (*Rizzolatti and Sinigaglia, 2016*).

As defined by *Rizzolatti and Sinigaglia, 2016*, p. 757, a mirror mechanism " (…) transforms sensory representations of others' behaviour into one's own motor or visceromotor representations concerning that behaviour.". It should be noted that the term 'mirror mechanism' is not uniformly understood (e.g. compare *Bonini et al., 2022* and *Rizzolatti and Sinigaglia, 2016*). We distinguish between a broad and a strict definition of a mirror mechanism. The broad definition states that the representation of an observed action (resp. the other's behavior) contains motor information about the observed action, but not sensory information. We define sensory information here as information about the environment outside the neural system that is exclusively of the type that exteroceptors, proprioceptors, and interoceptors can provide. From this we distinguish motor information that goes beyond sensory information and is unique to action control. This motor information comprises motor commands, defined as signals that determine which muscles are excited or inhibited and for how much and for how long, and components that are upstream of motor commands. Such upstream components are for example the goal or goal value of an action. The goal we define as an intended state of the environment. It is, in principle, possible that such upstream components are perceived by the subject or that they serve functions without the subject perceiving these components. Our strict definition of a mirror mechanism is more restrictive than the broad definition. It states that observed actions are represented as if the observer were performing them in place of the actor and that the representation is supramodal. This definition reflects the assumption that the primary purpose of a motor representation is the causation of an action. Accordingly, it is a representation from the viewpoint of the acting subject. This strict understanding of a mirror mechanism, which is arguably the most popular, underlay the design of this study. When we refer to the mirror mechanism in the following, we are referring to this strictly defined mirror mechanism.

The hypothesis that a mirror mechanism exists in the brain is closely related to two other hypotheses that build on each other and need to be clearly delineated: (1) The activity of F5 MNs during action observation is a motor representation of the observed action resulting from the mirror mechanism, here referred to as the 'representation hypothesis'. (2) The presence of such a motor representation of an observed action in F5 MNs is a prerequisite for a special kind of understanding of the observed action that cannot be achieved by purely sensory processing (named 'understanding from the inside' by *Rizzolatti and Sinigaglia, 2010*), here referred to as the 'understanding hypothesis'. There is currently insufficient evidence for or against the understanding hypothesis. To critically test this hypothesis, it would be necessary to operationalize 'understanding from the inside', experimentally manipulate F5 MNs, assuming they are a motor representation of the observed action, and measure the consequences for the 'understanding from the inside'. Fortunately, the representation hypothesis, the verification of which is a prerequisite for F5 MNs to have a function according to the understanding hypothesis, is easier to test. If the representation hypothesis were true, then the other´s action would have to be represented in the observer´s F5 MNs in the same way as the same action when performed by the observer. In other words, there would have to be a shared code for executed and observed actions in F5 MNs. Such a shared action code would be a necessary prerequiste for the

understanding hypothesis to be correct, yet, clearly falling short of being sufficient. However, the lack of evidence for a shared code would falsify the understanding hypothesis.

It may appear intuitive that the term 'mirror neuron' implies a shared action code. However, the definition of MNs by *Gallese et al., 1996* does not require a shared code (e.g., *Cook and Bird, 2013*; *Csibra, 2005*). In fact, the evidence for such a shared code is not fully settled and was limited for a long time to the work of *Gallese et al., 1996*. They concluded that "In most mirror neurons there was a clear relationship between the visual action they responded to and the motor response they coded." (p. 600). For more than 90% of the MNs, a so-called visual-motor congruence was reported, which could be strict (31.5%) or broad (60.9%). Common to all these neurons was that the actions studied were no better discriminated during observation than during execution and that the actions discriminated during observation were also discriminated during execution. However, the assessment of whether actions were discriminated in this study was based on a binary judgement: the presence or absence of a response in the sense of a discharge rate that differed from baseline levels. A comparison between observation and execution based on this criterion is an inadequate test for a shared action code because it does not account for potentially relevant differences in the discharge rates.

Doubts about the existence of a shared action code arise by subsequent studies that showed that the action-observation-related responses of many F5 MNs can be influenced by a number of factors like the spatial relationship of the observer to the observed action (*Caggiano et al., 2011*; *Caggiano et al., 2009*; *Maranesi et al., 2017*), and that in some MNs the direction of discharge rate modulation is opposite during execution and observation (*Kraskov et al., 2009*). These findings alone could be considered sufficient evidence against a shared action code and ask for alternative ideas on the function of F5 MNs to be considered (e.g., *Bonini, 2017*; *Csibra, 2007*; *Hsieh et al., 2010*; *Orban et al., 2021*; *Pomper et al., 2015*). However, this could be countered by the fact that in all these studies the results always applied only to a subset of the F5 MNs studied, and that the influence of the modulating factors mentioned could be so small compared to the influence of the action that the action is still coded similarly enough during execution and observation to be read out in the same way.

Given the lack of conclusive data on the existence of a shared action code, we set up this study which tried to clarify if a distinct group of F5 MNs that have a shared code exists and how well these MNs would discriminate observed actions. To this end, we designed an experiment that allowed us to compare responses to three types of actions during observation and execution. The actions differed in terms of several action variables that are known to have an influence on F5 MNs. These are 'grip type', 'manipulation type', and 'object target position'. The object was to be held in the target position to receive a reward, which is why the actions also differed in their action goal. Thus, a shared code should be detectable if it exists for at least one of these variables. To ensure that a shared code really refers to the actions, but not to the manipulated object itself, three objects of the same type were used. As reward is well known to influence responses it was kept constant (*Caggiano et al., 2012*; *Pomper et al., 2020*).

We examined if a shared code exists at single neuron and population levels. We found time periods with shared code in some neurons that can not be considered random. However, discrimination of observed actions with non-shared codes clearly prevailed and was optimal at times in the whole population. Thus, the hypothesis that F5 MNs represent observed actions as if the observer were performing them instead of the actor can only be maintained for a small group of F5 MNs under the additional assumption of a time-resolved readout. We discuss alternative concepts in light of recent findings and propose that F5 MNs represent goal pursuit from the observer's viewpoint. We highlight the possibility that it is not the goal pursuit of the observed actor that is represented, but the observer's own goal pursuit.

## Results

Two rhesus macaques were trained to perform the behavioral paradigm depicted in *Figure 1*. The paradigm consisted of two tasks that were performed in two separate blocks (*Figure 1A*): an execution task, in which the experimental monkey had to perform one out of three possible actions on objects with identical visual appearance (*Figure 1B*, top row), and an observation task, in which he had to observe the same actions carried out by a human actor in front of him (*Figure 1B*, bottom row). Each object was mechanically set up to allow only one of the three actions: it could be either lifted, twisted clockwise or shifted rightwards (from the viewpoint of the respective actor). Which action

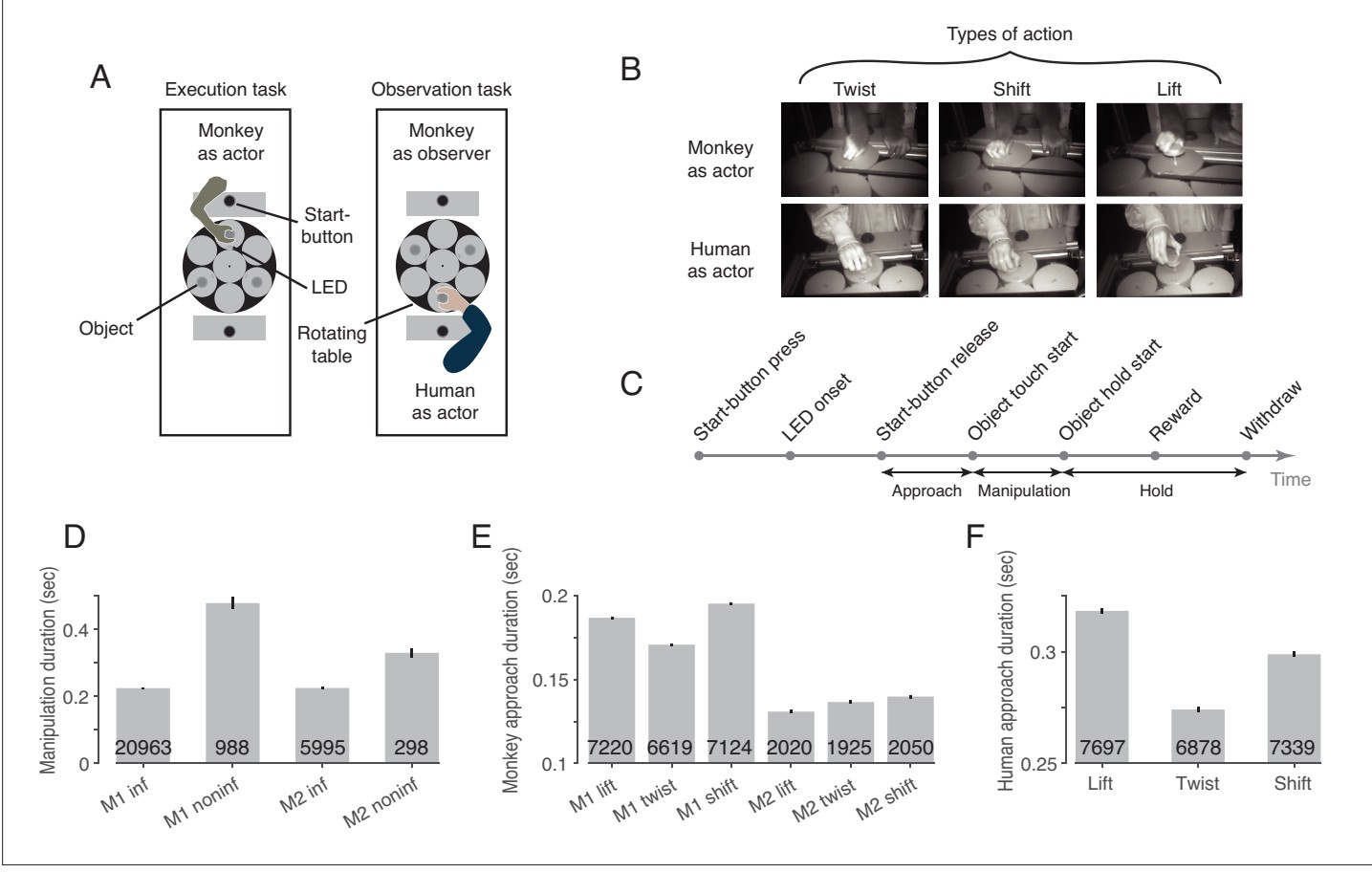

**Figure 1.** The behavioral paradigm. (**A**) Experimental setup with three identical objects positioned on a rotating table in front of the actor. Each object could be acted on in only one way. The type of action required was cued by the color of an LED next to the object. After LED onset, the actor was allowed to release the start-button. (**B**) Photographs of the three actions in execution and observation tasks at the time when the object was held in its target position. (**C**) The sequence of events in a trial. (**D**) Duration of manipulation epoch compared between trials with informative and non-informative cues (inf and noninf for monkey M1 and M2, error bars show 95% confidence intervals of the mean of all trials of sessions in which mirror neurons were recorded, pooled within a monkey, unpaired t-test, p<0.001 in each monkey, trial count per condition is indicated on the bars). (**E**) Duration of approach epoch compared between the three actions in trials with informative cue (error bars as in **D**, one-way ANOVA, p<0.001 in each monkey). (**F**) as **E** but for the human actor (p<0.001).

the object at stake afforded was indicated by a color cue provided by an LED next to the respective target object (*Figure 1A*). The target object was always the object coming to a hold right in front of the respective actor (the monkey or the human) after rotations (in darkness) of the table carrying the three objects. The directions and extent of the rotations varied across trials to prevent the monkey from predicting the action to be performed by resorting to knowledge of the arrangement of the objects and their associated actions. Distinct events allowed the separation of six epochs (*Figure 1C*).

To test whether the LED was actually used for action selection, we compared the duration of the manipulation epoch in trials in which the action cue was available with control trials in which an LED that did not contain information about the type of action was used instead of this informative LED. In trials lacking LED information, somatosensory feedback is required during manipulation of the object to identify the action that can be performed on the object, which takes time. Accordingly, we found a larger mean manipulation duration in trials with non-informative as compared to informative cues in both monkeys (*Figure 1D*). We additionally compared the duration of the approach epoch between the three actions in trials with informative cues, as we noticed that the reaching trajectory seemed to depend on the manipulation to be performed. For example, when shifting the object, a more arcuate movement was performed such that the object was touched on the left side and could be shifted more easily to the right. Consistent with this incidental observation, we found a difference in the

**Table 1.** The average duration (in ms) across trials of each epoch per action for monkeys and humans. The numbers in brackets indicate the lower and upper bound of the middle of 95% of data.

| | Monkey | | | Human | | |
|---|---|---|---|---|---|---|
| | twist | shift | lift | twist | shift | lift |
| Start-button press to LED onset | 3288 [1197, 4728] | 3608 [1242, 5211] | 3497 [1306, 4959] | 1272 [1029, 1507] | 1283 [1030, 1513] | 1277 [1029, 1510] |
| LED onset to Start-button release | 296 [224, 389] | 302 [231, 404] | 295 [221, 389] | 460 [335, 662] | 450 [324, 663] | 473 [337, 679] |
| Start-button release to touch | 163 [113, 219] | 183 [112, 248] | 175 [108, 243] | 274 [203, 409] | 299 [213, 422] | 318 [231, 461] |
| Touch to hold | 235 [140, 626] | 117 [59, 672] | 322 [199, 637] | 131 [73, 287] | 121 [60, 297] | 298 [183, 550] |
| Hold to reward | 566 [329, 804] | 570 [328, 805] | 568 [329, 803] | 570 [329, 806] | 569 [328, 804] | 570 [330, 804] |
| Reward to withdraw | 675 [296, 1325] | 744 [316, 1380] | 464 [288, 700] | 382 [229, 786] | 433 [276, 787] | 431 [257, 856] |

mean approach duration between the three actions in both monkeys (*Figure 1E*) and the human actor (*Figure 1F*).

To verify that the monkey actually observed the action of the human actor, we examined the monkey's gaze position in a 100ms window centered around the time when the actor touched the object. We found that the monkey's gaze was slightly offset from the object's center (horizontal offset (dva): –1.6±1.2 in lift, –1.7±1.2 in twist, –1.7±1.3 in shift [mean ± std across sessions of both monkeys]; vertical offset (dva): 0.2±0.9 in lift, 0.0±1.0 in twist, 0.0±1.0 in shift), but did not differ between the three actions (MANOVA: Wilks' Lambda = 0.98, $F_{(2, 375)}$=1.65, p=0.81). Key landmarks are the start position of the object (horizontal (dva): 0, vertical (dva): 0, measured at the center of the front top edge), its target position in lift (0, 2.3), twist (–1.6, 0.3) and shift (–2.2, 0), the position of start-button (0, 6.3) and the position of the two LEDs (±0.6,–2.3). Comparing the gaze position with these landmarks, it is evident that the monkey's gaze was directed to the area where the action was performed on the object.

We recorded 240 neurons from two monkeys, of which 177 (74%) met the mirror neuron criteria outlined in Material and methods and were used for the following analyses. As summarized in *Table 1*, the time between two adjacent events varied depending on the trial, action, and task. In order to make the timing of events comparable, we transformed the absolute times between events to a relative time by dividing the time interval between every two adjacent events into quartiles. Hence, each trial was divided into 24 time bins between the time of pressing the start-button and the time of withdrawing from the object.

## Single neuron analyses reveal a clear predominance of non-shared action codes, and a rare but not random occurrence of shared codes

Many, but not all, mirror neurons (MNs) discharged dependent on the type of action being observed. This dependence of the discharge rate on the observed action often existed only at certain times of the action. Moreover, for some neurons, the relationship between the discharge rate and the type of action observed, that is, the code for observed actions, also depended on time. This is exemplified by the neuron shown in *Figure 2A*. Focusing on the preferred action as a simplified description of the action code, we see that the neuron preferred a lift around the time of touch of the object but a twist and a shift during the later hold epoch. A dependence of action code on time bin is also evident for the neuron in *Figure 2B*. In this neuron, the discharge rate during action observation was highest for lift during the manipulation epoch, but highest for twist during the early hold epoch. Compared to the action code during observation, the action code during execution was temporarily the same only in a few neurons. For example, the neuron in *Figure 2A* preferred the same action during execution and observation except for the time bin before the onset of the hold epoch. However, in most neurons, time bins in which the action code during observation differed from that during execution

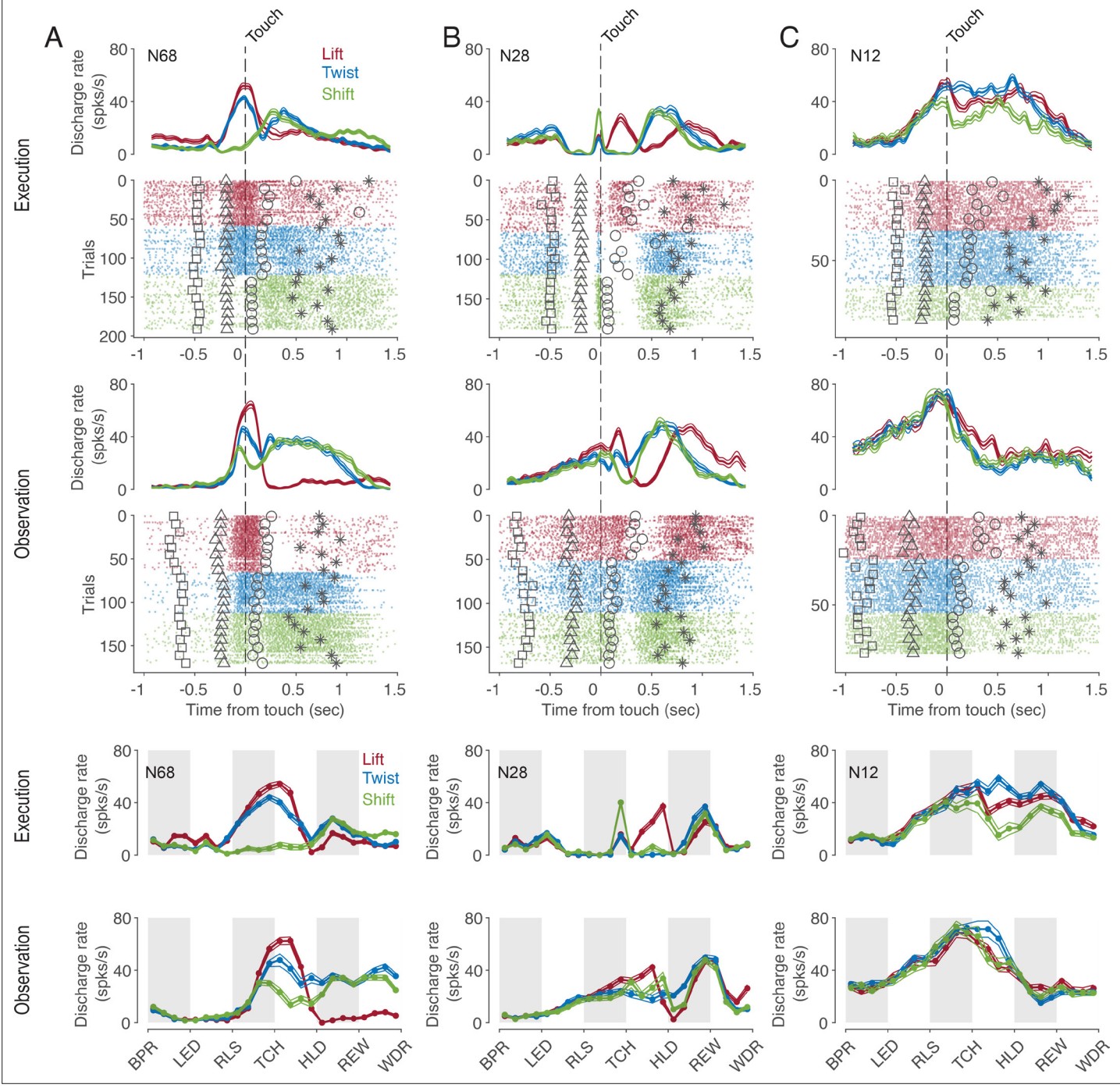

**Figure 2.** Exemplary F5 MNs. (**A**) A neuron that prefers a lift around the time of touch and then later a twist and a shift in both tasks. (**B**) A neuron that prefers a shift during execution and a lift during observation before the touch, a lift during manipulation in both tasks, and a twist after hold during observation. (**C**) A neuron with an action code during observation that differs from that during execution. In the upper panels, spike density functions (mean ± s.e.m.) and raster plots are aligned to the time of touch start (vertical dashed line), and markers indicate the time of four events around the time of touch: LED onset (square), start-button release (triangle), object hold start (circle), and reward (star). For better visualization, the events are shown for only about 10% of trials. In the lower two panels, discharge rate is plotted per relative time bin (vertical stripes, gray and white, indicate the six epochs). BPR: start-button press, LED: LED onset, RLS: start-button release, TCH: object touch start, HLD: object hold start, REW: reward, WDR: withdrawal, object release. N68, N28 and N12 are the IDs of MNs.

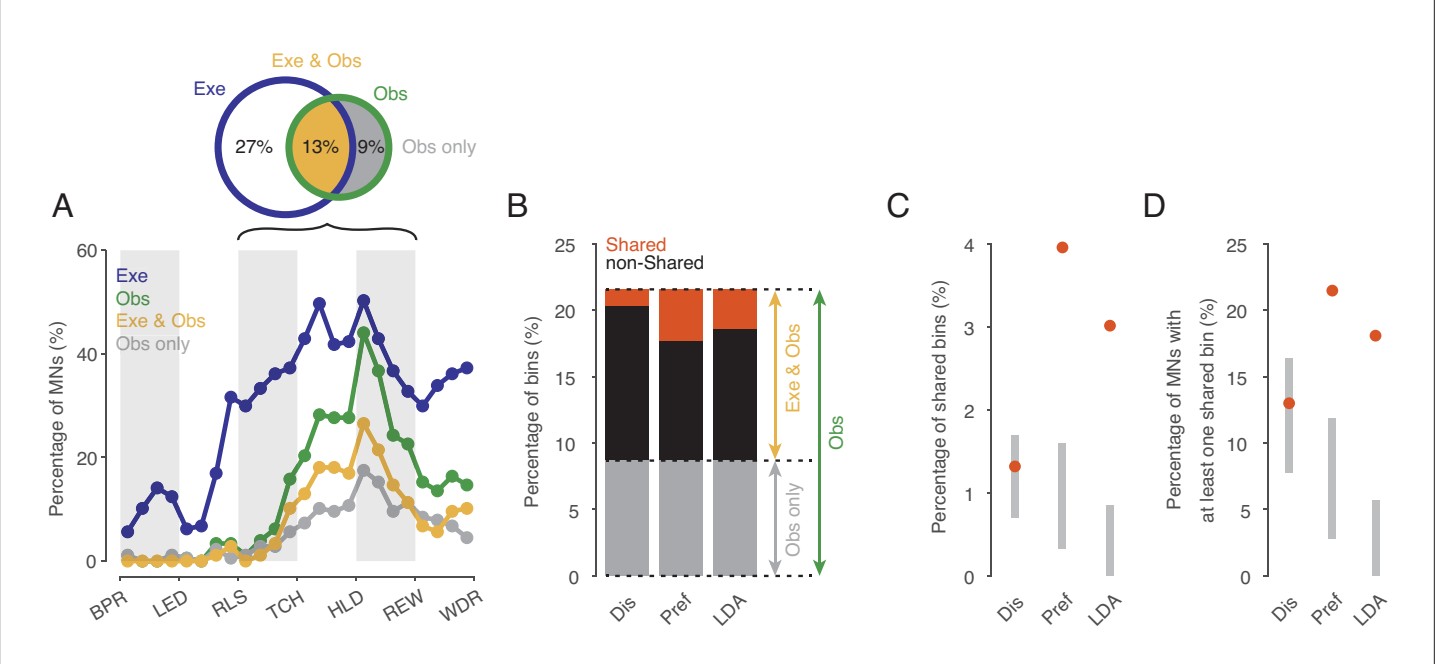

**Figure 3.** Shared action code in single neurons. (**A**) Percentages of MNs (n=177) discriminating between actions on execution (Exe), on observation (Obs), on both (Exe and Obs) and on observation only (Obs only) per time bin (Kruskal-Wallis tests, alpha <0.05 for Exe and for Obs, Benjamini-Hochberg-corrected for 177*24 tests). For the abbreviations of the events, see *Figure 2*. Black bracket indicates the action period consisting of three epochs further analyses focusses on. Venn diagram provides percentages of bins discriminating actions on Exe or Obs with respect to all bins examined in the action period (n=177*12). (**B**) Stacked bar chart showing the percentages of bins with respect to all bins examined in the action period with shared action code, non-shared action code and action code on observation only depending on the method used. Dis: same discharge method, Pref: same preference method, LDA: cross-task classification method using LDA. (**C**) Percentages of bins with shared action code with respect to all bins examined in the action period depending on the method (as in B). Gray bars indicate 95% confidence intervals derived from permutation tests under the null hypothesis that the coding of actions in the population of MNs at observation is independent of the coding at execution. (**D**) Percentages of MNs with at least one bin with shared action code related to all MNs (n=177). Methods and gray bars as in **C**.

predominated (as for example in the neuron in *Figure 2C* or in the time bin before touch in the neuron in *Figure 2B*) or in which actions were distinguished only during observation but not during execution (as for example in the time bin after hold in the neuron in *Figure 2B*).

To investigate the existence of an action code that is shared between execution and observation in a systematic manner we focused on the action period from release of the start-button to reward, which consists of 12 relative time bins. We first tested for each time bin and neuron whether actions were encoded at all when observed and when executed as indicated by the fact that the discharge rate depended on the type of action. The percentage of MNs encoding observed actions ranged from 0 to 45% depending on the time bin (green in *Figure 3A*). Starting with the second bin after release (RLS in *Figure 3A*), the proportion increased, reached its maximum of 45% in the first bin after reaching the target object position (HLD in *Figure 3A*), and decreased to about 20% shortly before the reward wasgiven. Of the total 2124 bins tested (12 bins * 177 MNs), 22% (n=458, *Figure 3A and B*) encoded observed actions.

In 60% of the n=458 bins that encoded the actions on observation, the actions were also encoded on execution (n=274, Exe & Obs, yellow in *Figure 3*). This means, conversely, that 40% of the bins with action coding during observation did not exhibit any action coding during execution and must therefore be eliminated as carrier of a shared action code (Obs only, gray in *Figure 3*).

The bins in which observed actions were discriminated were distributed across all 177 MNs examined such that 72% of MNs (n=128, 53% of F5 neurons examined) encoded the observed actions in at least one bin. In both observation and execution, the actions were encoded in at least one bin by 49% of MNs (n=87, 36% of F5 neurons). Another 19% of MNs (n=33, 14% of F5 neurons) encoded the actions on observation in at least one bin and also on execution in at least one bin, but without containing a bin in which the actions were encoded during both observation and execution. Only 5%

of MNs (n=8, 3% of F5 neurons), encoded the actions on observation in at least one bin but not on execution in any of the 12 bins of the action period. In summary, more than two thirds of the MNs encoded the observed actions in at least one bin and more than 90% of them encoded the actions on execution in at least one bin, although the bins did not have to be the same.

For the bins in which actions were encoded when observed and when executed (n=274, indicated yellow in *Figure 3B*), we tested if the action code was shared. To this end, we used three alternative approaches. The first approach (referred to as 'same discharge method', Dis in *Figure 3*) aimed at testing for a 'perfect' shared code. The shared code is 'perfect' if the discharge rates associated with each action are the same during execution and observation (for more details see Material and methods). Of the bins examined (n=274), about 10% (n=28) exhibited such a shared code (*Figure 3B*). In relation to all bins in which the observed actions were encoded (n=458), 6% had a shared code, but 94% did not. The proportion of bins with a shared code in relation to all bins examined (n=2124) was 1.3% (*Figure 3C*). To assess whether this proportion is large enough to infer physiological significance of these shared bins, we compared it with the proportion of bins with shared code that would be expected by chance if the coding of actions in the population of MNs at observation were independent of the coding at execution (for more details see Material and methods). Testing against this null hypothesis was first proposed by *Csibra, 2005*. It turned out that the proportion found does not differ significantly from the proportion predicted by this null hypothesis (*Figure 3C*). The qualitatively same result was obtained when considering the proportion of MNs that had a shared code in at least one bin. This proportion was 13% (n=23, 10% of F5 neurons), which is also in a range to be expected by chance (*Figure 3D*).

The second approach (referred to as 'same preference method', Pref in *Figure 3*) uses a less stringent criterion for same action coding. Here we considered only whether the action that had the highest or lowest discharge rate when executed also had the highest or lowest discharge rate when observed and concluded on a shared code if this was the case (for more details see Materials and methods). This criterion leans on the idea that a neuron responds to only one action. Using this method, of the bins examined (n=274), 31% (n=84) had a shared code (*Figure 3B*). In relation to all bins in which the actions were encoded during observation (n=458), 18% had a shared code, but 82% did not. The proportion of bins with a shared code in relation to all bins examined (n=2124) was 4.0% (*Figure 3C*). This proportion is higher than to be expected by chance according to the aforementioned null hypothesis. The proportion of MNs with a shared code in at least one bin was 21% (n=38, 16% of the F5 neurons), which is also above the range to be expected by chance (*Figure 3D*).

The third approach relies on linear discriminant analysis (referred to as 'cross-task classification method', LDA in *Figure 3*). We tested whether the observed actions could be predicted significantly better than chance by a linear classifier trained on trials during execution (for further details see Materials and methods). In contrast to the same preference method, the discharge rate is used here without any information about whether an action is being executed or observed (since neither offset correction nor rescaling or ranking is used). In contrast to the same discharge method, the discharge rate can vary in the mean and in the distribution (across trials) between the actions. Of the bins examined (n=274), 23% (n=64) turned out to have a shared code (*Figure 3C*). In relation to all bins in which the actions were encoded during observation (n=458), 14% had a shared code, but 86% did not. The proportion of bins with a shared code in relation to all bins examined (n=2124) was 3.0% (*Figure 3D*). This proportion is higher than would be expected by chance according to the aforementioned null hypothesis. The proportion of MNs with a shared code in at least one bin was 18% (n=32, 13% of the F5 neurons), which is also above the range to be expected by chance (*Figure 3E*).

Hence, all three methods document that the overwhelming majority (about 80–95%) of bins in which action-related information is encoded when observed do not have a shared code. However, both the 'same preference' method and the 'cross-task classification' method also show that the proportion of bins that exhibit a shared code is higher than would be expected if the encoding of actions in the population of MNs during observation were independent of the encoding during execution. This suggests that at least certain aspects associated with the observed action are coded in some MNs in the same way as when the monkey performs the action himself.

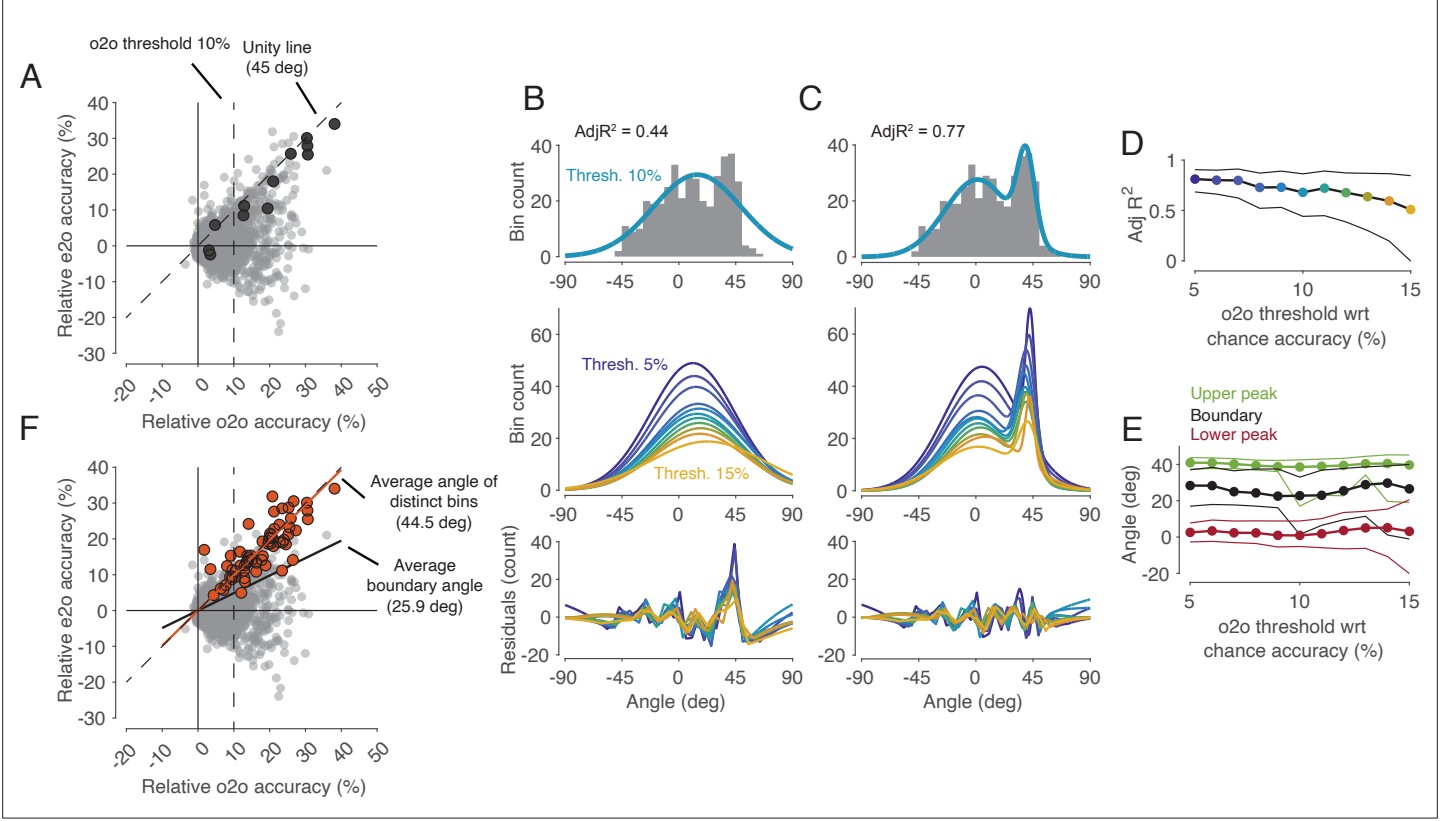

**Figure 4.** Comparison of classification accuracy for observed actions of classifiers trained either on execution (e2o) or on observation (o2o). (**A**) Each dot in the scatter plot represents the e2o accuracy and the o2o accuracy in a time bin of a neuron, relative to chance level. Each neuron contributed to this plot with 12 time bins between start-button release and reward. The black dots indicate the data points of an example neuron (N68). Threshold 10% is one instance of the 11 filtering thresholds. (**B**) Top: The gray histogram shows the distribution of the angles of each dot above threshold 10% in **A**. The envelope depicts the single-Gaussian fit. Middle: single-Gaussian fits for 11 different thresholds. Bottom: The residuals of the single-Gaussian fits for the 11 thresholds. (**C**) : The two-Gaussian fit for threshold 10%. Middle and bottom: two-Gaussian fits and residuals as in **B**. (**D**) Adjusted R-squared (AdjR$^2$) of two-Gaussian fits for thresholds 5 to 15%. Colored dots indicate the means and the black lines the 95% confidence intervals across 1000 bootstrapped resamples. (**E**) Upper peak, lower peak, and boundary (trough between the two peaks) of two-Gaussian fits for thresholds 5 to 15%. Green, black, and red dots indicate the means and the lines the 95% confidence intervals across 1000 bootstrapped resamples. (**F**) Same scatter plot as in **A**. The orange dots indicate the bins with significant e2o accuracies and the orange line shows their average angle. The average boundary angle (black line) corresponds to the mean boundary angle across the 11 thresholds in **E**.

## There is a distinct cluster of time bins with a shared code according to the representation hypothesis

In the cross-task classification method, the criterion used to decide whether a shared code exists is that observed actions can be predicted significantly better than chance by a classifier trained on trials during execution. However, this criterion is too weak to test the representation hypothesis. In fact, to accept the existence of a shared code where in F5 MNs observed actions are encoded according to their motor representation, it is required that the classifier trained during execution is the best one to read out observed actions. In other words, the accuracy of a classifier trained on observation should not be better than the accuracy of the classifier trained on execution. To compare the two accuracies for all bins (n=2124), we plotted the accuracies of predicting the observed actions by classifiers trained on execution (execution-to-observation, e2o classifier) against that classifiers trained on observation (observation-to-observation, o2o classifier, *Figure 4C*). This comparison revealed a small but clearly distinguishable cluster of bins in which the accuracy was nearly the same for the two types of classifiers (4B-E). Almost all the previously mentioned bins that had a significant accuracy of observed actions with a e2o classifier (n=64) fell into this group (*Figure 4F*). The demonstration of this definable cluster of bins supports the representation hypothesis for F5 MNs that contain such time bins.

## Mirror neurons with exclusively shared code time bins are extremely rare

Following the idea that single neurons can be ascribed the property of representing observed and executed action aspects in the same way, we examined whether there are 'pure mirror neurons', defined as neurons that encode observed actions exclusively with the code used during execution. To this end, we considered the distribution of bins with a shared code and bins in which observed actions were encoded with a non-shared code or not encoded at all during execution (*Figure 5A*). The bins with a shared code (n=84 for the same preference method, n=64 for the cross-task classification method) were contributed by 38 and 32 neurons, respectively. Only three (Pref method) or two (LDA method) of these neurons exhibited exclusively shared bins, and the neurons differed depending on the method (IDs 44, 154, 96 for Pref, IDs 9, 84 for LDA). This means that almost all MNs that encode observed actions with a shared code do so only at certain times, but encode the actions differently at other times. A 'pure mirror neuron' is an exception. The property of encoding observed actions with a shared code can thus be described as a dynamic property of a subset of MNs (21% and 18% of MNs, 16% and 13% of F5 neurons, for same preference and cross-task classification method, respectively).

## Action segments of different durations are coded, without major differences in relative frequency between action segments with shared and non-shared code

If there are aspects of action that are coded in the same way, then the question arises as to what these are and whether they differ from those that are coded in a different way. To address this question, we compared the frequency distribution of action segments with shared code with the frequency distribution of action segments with non-shared code (*Figure 5B–E*). –We defined an action segment as the set of consecutive bins with the same type of code, shared or non-shared, adding to the non-shared code segments the bins in which the action was encoded only when observed but not when executed ('obs only' in *Figure 5A*). An action segment starts with a bin in which the observed action is encoded (shared or non-shared), and ends before a bin in which the observed action is encoded either by the other code type or not at all. With 12 time bins, a segment can have a duration between 1 and 12 bins and a starting position between 1 and 12. This results in 78 possible segments that can be coded (*Figure 5B*, bottom). For both the same preference and the cross-task classification method, the absolute frequencies showed that for almost all occurring action segments, there were more often those with non-shared code than with shared code (*Figure 5B*). This result is not unexpected in view of the overall predominance of bins with non-shared code described above. For both types of segments, shared and non-shared, it can be noted in *Figure 5B* that (1) segments of shorter duration predominated, (2) segments with a duration of only one bin after reaching the target position of the object were particularly frequent, (3) there were hardly any action segments with a duration of 7 or more bins. There were two MNs that appeared to encode the entire action, from a bin before reaching the object to the reward (IDs 64 and 19). However, inspection of these two neurons revealed that although the neurons distinguished the observed actions throughout the action, the action code changed. Notably, both in these two neurons and in the neurons with segments consisting of 7 bins, the encoding changed once the object had reached its target position.

The comparison of the relative frequencies of action segments showed only minor differences between the segments with shared and non-shared codes (*Figure 5C*). One of these differences was the dominance of shared segments with a duration of just one bin in the period starting with the hand having reached the object. Note that shared segments with only one bin should not be considered random, as they were more frequent than would be expected according to the aforementioned null hypothesis (*Figure 5D*). The onset of segments, regardless of their duration, differed between shared and non-shared mainly in that non-shared segments occurred earlier in the approach epoch than shared segments (*Figure 5E*, left). The relative frequency of segment duration did not differ between shared and non-shared segments (*Figure 5E*, right). In summary, this descriptive analysis showed that (1) not the entire observed action, but action segments are encoded, (2) reaching the target position plays a special role in this, (3) shared and non-shared coded segments hardly differ with respect to starting point and duration, with the exceptions that (4) non-shared segments occur earlier in the approach epoch, and (5) shared segments occur more often after the hand has reached the object.

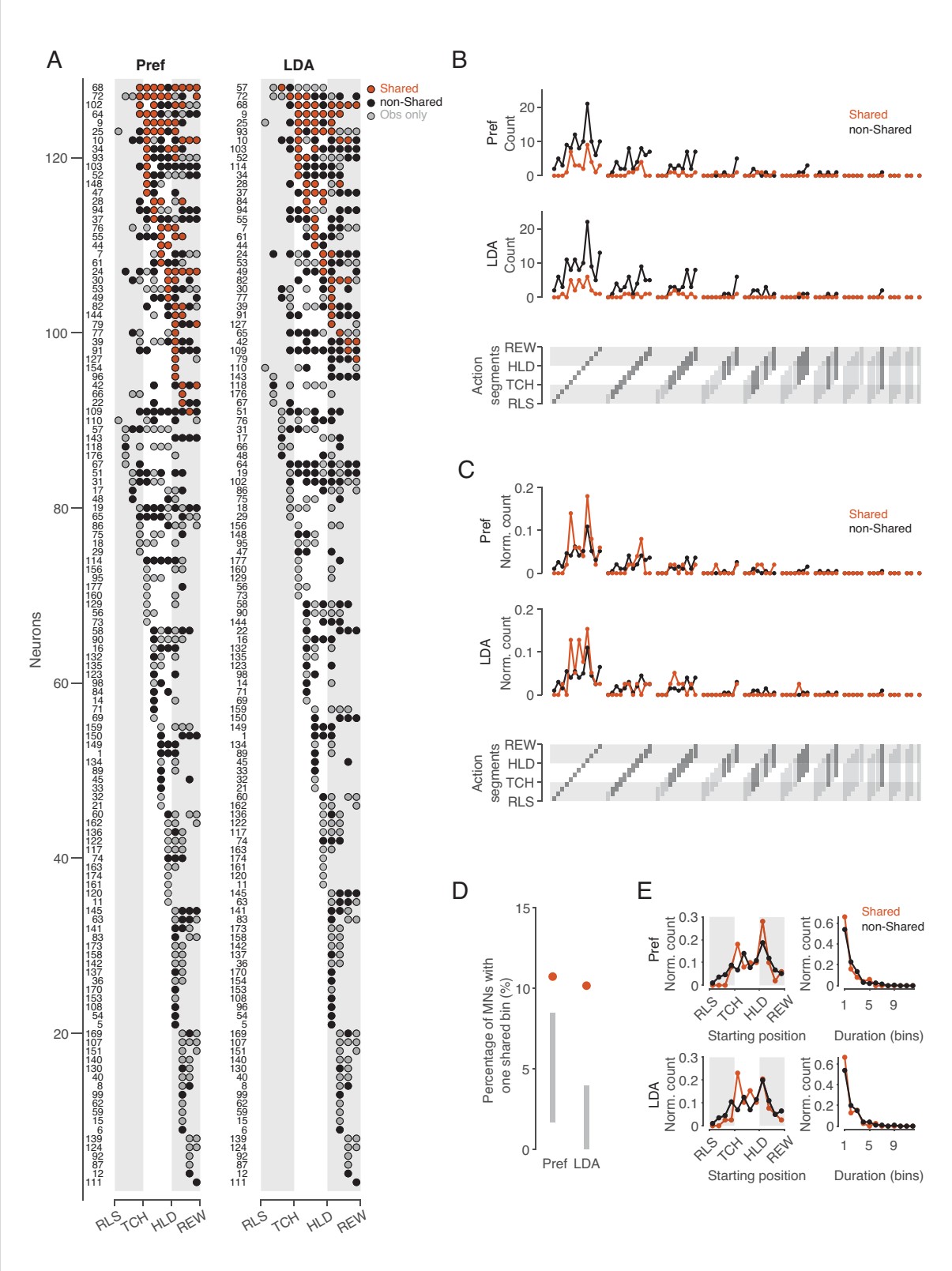

**Figure 5.** Distribution of shared and non-shared codes across time bins and neurons. (**A**) The bins in which observed actions were encoded are shown for all neurons with at least one such bin (rows) across relative time (columns). Bins are classified into bins with shared code, bins with non-shared code, and bins that do not distinguish actions at all when executed (obs only). Neurons are ordered by (1) occurrence of at least one shared vs. no shared bin, (2) first occurrence of shared bin, (3) number of shared bins, (4) first occurrence of non-shared or obs only bin, (5) number of non-shared or obs

*Figure 5 continued on next page*

*Figure 5 continued*

only bins. Left: action preference method, Right: cross-task classification method using LDA. Numbers to the left indicate the ID of each neuron. For the abbreviations of the events, see *Figure 2*. (**B**) Count of action segments of consecutive bins with shared or non-shared (including obs only) codes separately for the two methods (top and middle). Segments indicated by light gray did not occur (bottom). (**C**) As B, but count of action segments normalized to the total number of bins of one code type. (**D**) Percentages of MNs with segments consisting of one bin with a shared code. Methods and gray bars as in *Figure 3D*. (**E**) Normalized count of segments depending on starting position (left) and duration (right), separately for the two methods.

## Subpopulations with a shared code can be identified, but discriminate actions worse than the whole population with a non-shared code

We may safely assume that populations of cortical neurons, rather than single neurons, determine an animal's behavior or perception. Hence, we asked whether the whole population of MNs or a subpopulation thereof codes observed actions with a shared code. To address this question, we first considered the whole population of all MNs studied (n=177). The mean accuracies of different classifiers based on the single neuron analysis described earlier are shown in *Figure 6A*. It is confirmed that (1) observed actions were discriminated later by execution-trained classifiers (red in *Figure 6*) than by observation-trained classifiers (green), and (2) observed actions were discriminated best by observation-trained classifiers after reaching the target position of the object. A result not yet described was that observed actions were discriminated as well as executed actions at this time bin, as evidenced by the fact that the mean accuracies of the e2e classifier (blue) and the o2o classifier did not differ. The mean accuracy of the classifiers, each based on the discharge rate of single neurons, was comparatively low with a maximum of 43% (chance 33%). This is explained by the fact that a large proportion of neurons did not discriminate actions at all in a given time bin (see *Figure 3A*) and that neurons that did discriminate actions did not discriminate them completely (see *Figure 4A*). If, in contrast, one trains classifiers not on the discharge rate of single neurons but on the discharge rate vector formed by the discharge rate of all 177 MNs, one bypasses the limitation of single neurons and suppresses the effect of those neurons that do not discriminate actions. Accordingly, for both types of within-task classifiers, e2e and o2o, higher accuracies were achieved with an optimal value of 100% in the bin where the target object position was reached (*Figure 6B*). As expected, the accuracies of the e2o (and also o2e) classifiers were also higher than the mean accuracy of these cross-task classifiers trained on single neurons. However, with maximum values of 50–60%, the accuracy was still lower than the accuracy of the within-task classifiers. If one applies the criterion for a shared code, as required by the representation hypothesis, that observed actions may not be better discriminated with classifiers trained by observation than with classifiers trained by execution, then actions in the whole population are not shared coded.

This leaves open the question whether a subpopulation of MNs exists that has accuracies for e2o classifiers that are not worse than the accuracies of the o2o classifiers of this subpopulation and that are also not worse than the accuracies of the o2o classifiers of the whole population of F5 MNs. Neurons eligible for such a subpopulation are those classified as MNs with at least one bin with a shared code using the cross-task classification method at the single-cell level. However, these MNs often had non-shared codes in other bins that could counteract a shared population code in those bins. Accordingly, the population analysis of such a selected subpopulation (n=32, neurons with bins with a shared code in *Figure 5A*, right) showed a higher e2o accuracy than the total population (~65%), but in most bins a worse e2o accuracy compared to the o2o accuracy (*Figure 6C*). Alternatively, one might assume that only MNs that 'clearly' encode the observed actions constitute a subpopulation with shared code. An artificial threshold for o2o accuracy can be set, whose crossing is considered a clear encoding. We set a threshold of 10% above chance level for o2o accuracy (see the threshold plotted *Figure 4A*, bins were included without any statistical test on single-cell level). Depending on whether the e2o accuracy was above or below the average boundary angle (see *Figure 4F*), we classified a bin as shared or non-shared (referred to as 'threshold method'). Restricting the consideration to neurons that contained only shared bins (n=17) avoided the inclusion of neurons that contained relevant non-shared bins. As a result, the population analysis showed, as expected, a good agreement of the e2o and o2o accuracies in the action period (*Figure 6D*), that is, a shared code. However, these accuracies were in the range of 50–60%, which corresponded to the e2o accuracy of the whole population. Thus, although observed actions in this subpopulation were not discriminated worse with execution-trained classifiers than with

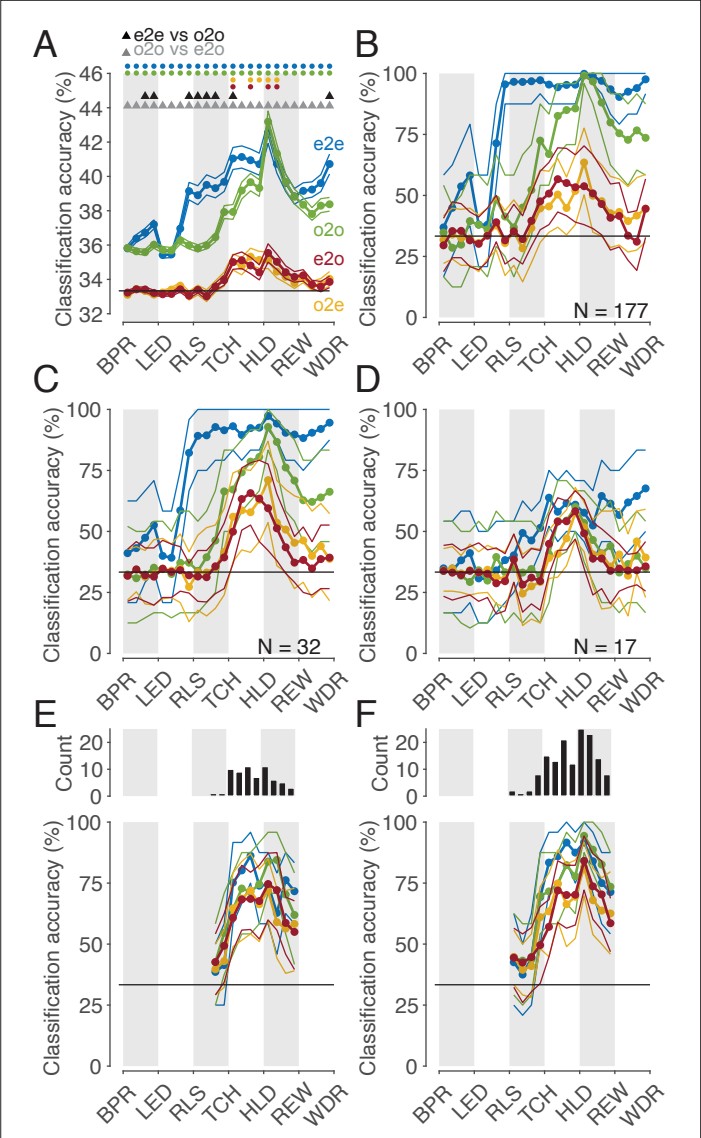

**Figure 6.** Classification of the three actions. (**A**) Single neuron classification accuracy (mean ± s.e.m, n=177 neurons) per time bin, execution-trained classifiers tested with execution trials (e2e, blue), tested with observation trials (e2o, red). Observation-trained classifiers tested with observation trials (o2o, green), and execution trials (o2e, orange). Colored dots indicate time bins of the same color with significant accuracy above chance (Benjamini-Hochberg-corrected one-sided signed-rank tests, alpha <0.05). Triangles indicate time bins with significant differences in the accuracy between the indicated classifications (Benjamini-Hochberg-corrected two-sided signed-rank tests, alpha <0.05). For the abbreviations of the events, see *Figure 2*. The very low but above chance level accuracy of the e2e and o2o classifiers before LED onset indicates that the three conditions were already distinguishable to some extent before LED onset by cues we could not identify. (**B**) Population classification accuracy per time bin (mean and 90% CI derived from bootstrapping). Same as **A**, but here, all neurons (n=177) constructed the 177 features of a classifier. (**C**) Same as **B**, but the population consists of only 32 neurons with at least one bin with a shared code according to *Figure 5A*, right. (**D**) Same as **B**, but the population consists of only 17 neurons with only bins with a shared code according to a 10% threshold criterion and the boundary angle shown in *Figure 4F*. (**E and F**) Same as **C** and **D**, respectively, but for each time bin only neurons with a shared code in this bin were included, which leads to a variable population size (top), and in F, not only neurons with only bins with a shared code (as in **D**), but for each time bin all neurons with a shared code in this bin were included.

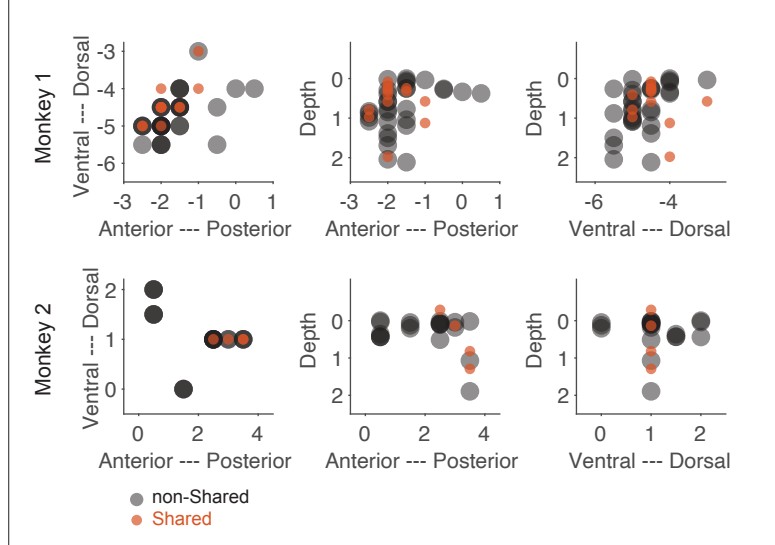

**Figure 7.** Anatomical location of neurons with only shared bins and neurons with only non-shared bins selected based on the threshold method (see main text). Each dot represents the recording site of a neuron. Dark colors indicate overlap of neuron locations. The numbers of the axes refer to the electrode position (in mm) in relation to the center of the recording chamber and to the cortical surface (for depth).

observation-trained classifiers, they were discriminated considerably worse than with observation-trained classifiers in the whole population.

The question notwithstanding how this could be achieved, one might assume that there are target structures, such as populations of downstream neurons, that read out action-related information only at specific times from specific MNs with shared codes. To estimate what accuracies could be achieved under this assumption we selected neurons bin-wise for both selection procedures described before. A neuron was selected for population analysis in a time bin if it had a shared code in that bin (6 E, F, top). We found e2o population accuracies of up to 80% (*Figure 6E and F*). The accuracies were higher for the second selection procedure, which could be due to the higher number of included neurons. For both procedures, the e2o accuracies were slightly below the o2o accuracies, and the o2o accuracies were below the o2o accuracies of the whole population (*Figure 6B*). This indicates that even in subpopulations of neurons selected in the best possible way, the observed actions are not coded as well with a shared code as with a non-shared code in the whole population.

Despite the low number of neurons with a shared action code, they might still serve a significant function. One indication of such a distinct functionality could be an anatomical separation between neurons consisting of only shared bins and those consisting of only non-shared bins. As shown in *Figure 7*, the anatomical locations of these two subpopulations were largely overlapped. Hence, we did not find convincing evidence that neurons with only shared bins are anatomically separated from neurons with only non-shared bins.

## Discussion

In summary, we showed that three observed actions performed on visually identical objects and differing in the required grip type, manipulation type, and the object's target position were well discriminated in the whole population of F5 MNs studied. Discrimination began during approach to the object, at a time when the actions became discriminable, and it was optimal when the object's target position had been reached. While the action-specific activity states of the whole population during observation were – to some extent – similar to the states during the execution of corresponding actions, they were not similar enough to allow the observed actions to be decoded as well by linear classifiers trained on action execution as by those trained on action observation. In contrast to the demonstration of such non-shared population codes in the whole population, analyses of the single neurons revealed that there were time bins in up to about 20% of MNs in which observed actions were

discriminated equally well by observation- and execution-trained classifiers. While these time bins with shared code were rare and co-occurred with non-shared code bins in most neurons, they formed a distinct cluster and are not to be considered random. Population analyses on subsets of neurons containing such bins with shared code revealed that observed actions could be discriminated equally well by observation-trained and execution-trained classifiers when neurons were selected bin-wise. However, the accuracies of such shared population codes for subpopulations were worse than those of the non-shared population codes for the whole population.

Our finding of a low proportion of MNs with shared codes is similar to the finding of a proportion of 5.5% 'strictly congruent' MNs relative to the F5 neurons studied (29/531) reported by *Gallese et al., 1996*. In our study, depending on the criteria used, the proportion of MNs with shared code relative to all F5 neurons ranged from 0.8% to 16% (from 2/240 MNs with only bins with a shared code using the cross-task classification method up to 38/240 MNs with at least one bin with a shared code using the preference method). However, the crucial difference to Gallese et al. is that we found that the proportion of 'incongruent' neurons, that is, neurons without bins with shared coding, is not below 10%, but forms the clear majority. Moreover, most MNs that are classifiable as 'congruent' based on the occurrence of at least one bin with a shared code also contain bins with a non-shared code. This explains why the whole population encodes the observed actions very well with a non-shared code. The discrepancy between our result and that of Gallese et al. is probably explained by the fact that we did not compare the action-specific responses of a neuron but the action-specific discharge rates of a neuron between execution and observation. Our finding that MNs discriminate observed actions well by non-shared codes is in line with a recent report by Papadourakis and Raos (2019), who tested the coding of actions that differed in the type of grip afforded by the object being grasped. The proportion of F5 MNs with the same grip preference for action execution and observation was so small that they were considered random, at odds with our finding of a small, yet non-random subset of MNs with at least one bin with the same action preference (see *Figure 3D*). One explanation for this discrepancy would be the way they assigned the preferred grip to a neuron based on the highest discharge rate without using a statistical test. This explanation is supported by the results of their and our population analyses, each of which found a low but above chance level similarity of action-specific activity states. In their case this was shown by a population representational similarity analysis, in our case by the cross-task population classification (see *Figure 6B*, red curve).The ability of premotor MNs to discriminate grasping actions very well during both observation and execution is also evident from the study by *Mazurek et al., 2018* (see their Figure 3). With regard to shared action coding, this study determined a similarity index per neuron. This index showed a broad distribution, with the proportion of neurons with a shared code appearing to be higher than ours (see their *Figure 3*). However, this proportion is not directly comparable to ours because no statistical test was performed at the level of single neurons and the similarity index used has the flaw of pretending a shared code when the actions are not distinguished at all (see our Materials and methods for more details). In summary, the results of the two studies are not inconsistent with our finding that (1) in the single neuron analyses, non-shared codes clearly predominated, but some MNs contained periods of shared codes that cannot be considered random, (2) in the population analyses, actions were better distinguished by non-shared codes of the whole population than by shared codes of differently selected subpopulations.

## Implications for 'strict mirroring' in F5 mirror neurons

Our finding is evidence against the representation hypothesis, which is based on a strictly defined mirror mechanism. That is to say, the activity of F5 MNs, as defined by *Gallese et al., 1996*, during action observation is not a supramodal motor representation of the observed action, as if the observer were performing it in place of the actor. At best, the representation hypothesis might hold for a subset of F5 MNs, yet only for restricted segments of an action (see below). Therefore, the prerequisite for the understanding hypothesis is not met for the whole population of F5 MNs. The objection could be raised that it is sufficient for the representation hypothesis to hold that observed actions could be decoded from the whole population with execution-trained classifiers with an accuracy that is above chance level (see *Figure 6B*). This finding indeed indicates that there is a similarity in the action-specific activity states of the whole population between execution and observation, a similarity which in principle could be exploited by a biologically plausible readout. However, since the observed actions were much better distinguished with observation-trained classifiers, the action-specific activity

during observation must be more than a supramodal motor representation of the observed action. Therefore, the representation hypothesis for the whole population of F5 MNs must be rejected.

In contrast to the whole population of F5 MNs, about 20% of the MNs in our study that contained time bins with shared code met the requirement for a motor representation according to the strictly defined mirror mechanism in these time bins. If this information could be selectively read out from these MNs, taking time into account, these MNs could in principle be the substrate of sparse coding of observed actions according to the mirror mechanism (*Földiak, 2002*; *Vinje and Gallant, 2000*). That F5 MNs may comprise functionally different subsets of MNs with some serving shared coding at particular times may not be implausible given that they differ in their anatomic location and connections (*Kraskov et al., 2009*) as well as their cellular identity (*Ferroni et al., 2021*). Consistent with this, F5 MNs differ in how they modulate during the execution or observation of actions and whether or not certain factors influence them (*Caggiano et al., 2011*; *Maranesi et al., 2013*). Whether some MNs at some time points indeed function to provide motor information about the observed action according to the strictly defined mirror mechanism cannot be conclusively answered at present. However, the following three findings rather argue against it. First, simulating decoders that time-dependently read from the population of MNs with shared code did not achieve the performance of decoders that read from the whole population. Second, the comparison of the duration and starting position of the segments with shared and non-shared codes did not provide convincing evidence that segments with shared code represent specific action aspects. Third, there is as yet no evidence that the neurons of such subpopulations act together on a target. For example, we could not find convincing evidence for a specific anatomical location of the neurons of a subpopulation with only shared codes compared to those with only non-shared codes.

The fact is that the overwhelming majority of F5 MNs encoded observed actions in a non-shared manner at most times. And there are reasonable doubts whether the shared coding that occurs at times in some neurons really has a function according to the strictly defined mirror mechanism. Therefore, one may wonder what other purpose the observation-related responses of the F5 MNs might serve. In what follows, we distinguish between four alternative concepts and critically compare them with our results and previous findings.

## Alternative concept 1: sensory concept

The first alternative concept posits that F5 MNs' activity is a sensory representation of aspects of the environment that could be relevant for the subject's own action control, that is, for action selection, planning, preparation, and execution. This can be, for example, the appearance of an object or a change in the own finger configuration caused by the execution of the own action. But also the action of the other is a change in the environment. A sensory representation of expedient information is based on sensory input and/or predictions. The crucial difference we make here compared to a motor representation is that the information about the environment is exclusively of the kind that exteroceptors, proprioceptors and interoceptors can provide. Thus, it does not include information that we consider unique to action control, such as information about a goal, here defined as an intended state of the environment, or information about motor commands, here defined as which muscles to excite or inhibit, how much, and for how long. According to this concept, the predominance of non-shared codes in our study may be a consequence of the fact that for discrimination of actions only somatosensory input was available during execution, whereas only visual input was available during observation. This explanation assumes a multimodal sensory representation. However, if we consider a supramodal sensory representation, then non-shared codes can be explained by, from the monkey's viewpoint, different target positions of the object during execution and observation. For example, if the actor shifts the object to the right from his point of view, the observer sitting opposite of him observes a shift of the object to the left. The occurrence of shared codes could be led back to similarities between the represented aspects of the actions during execution and observation such as the upward position of the target object relative to the table. Thus, our results can in principle be explained by the concept of sensory representation. However, evidence from previous studies suggests otherwise. In terms of action execution, neural responses of F5 neurons to a visual stimulus such as the presentation of an object have been shown to represent not the sensory properties of the stimulus itself, but the possible or upcoming action, for example, the way of grasping the object afforded by the object properties (*Schaffelhofer and Scherberger, 2016*). With respect to action observation, findings that

many F5 MNs are influenced by the observer's reachability to the observed action (*Caggiano et al., 2009*) and by the expected value of the goal for the observer (*Pomper et al., 2020*) show that the activity of many F5 MNs cannot be explained by this 'sensory concept'.

## Alternative concept 2: affordance concept

The second alternative concept posits that observation-related activity of F5 MNs is a motor representation, yet not one of the observed action according to a mirror mechanism. Instead, the concept posits that the activity reflects possible actions by the observer in response to the observed action that might be performed under natural conditions (*Pomper et al., 2015*). Following Gibson, these action options can be referred to as 'social affordances' (*Gibson, 1979*; *Orban et al., 2021*), and according to the 'affordance competition hypothesis', multiple action options could be represented simultaneously, competing with each other for execution, influenced by factors like the expected reward associated with each action option (*Cisek, 2007*). This concept is supported by the findings that the activity of F5 MNs depends on whether the observer can reach the observed action (*Caggiano et al., 2009*) and how much reward the observer can expect after the action (*Caggiano et al., 2012*; *Pomper et al., 2020*). It is to be expected that possible responses to the observed action will hardly ever be pantomimes of the observed action, which might explain our finding that non-shared codes prevailed. The occurrence of shared codes, albeit rare, remains hard to explain. An even bigger challenge for this concept arises if we consider the fact that there was no way for the observing monkey in our experiment to execute any of the action options. Therefore, it seems ecologically inefficient to represent specific response options for each of the three observed actions, which, moreover, would have to be assumed to change due to the modulation of the neurons in the course of an observed action. Hence, this 'affordance concept' does not offer a good explanation either.

## Alternative concept 3: goal-pursuit-by-observer concept

The third alternative concept posits that the activity of F5 MNs during observation is a motor representation of an 'action' by the observer, with the observer himself pursuing a goal. By 'goal' we refer to an intended state of the environment. A state is intended because its occurrence is associated with a value for the subject – the 'goal value'. In our experiment, the state that has direct value for the observer is the receipt of water following the observation of the action of the other. Therefore, we consider the receipt of water to be the primary goal of the observer. For the human actor, we deem the primary goal to be hearing the click sound associated with the delivery of water to the observer, signaling the successful execution of the action, which has value for the human actor. This click sound, or the receipt of water, is the last event in a series of events that can be conceptualized as goal pursuit. The pursuit of a goal is a process that begins with the setting of a goal and, if successful, ends with the achievement of the goal. Goal pursuit consists of a sequence of intermediate goals, some of which usually require specific muscle activation patterns controlled by motor commands. In our experiment, we consider releasing the hand from the start button, touching the object, reaching the target position, and holding the object in the target position as intermediate goals. We call the last of these intermediate goals, which precedes the primary goal and which, in the case of one's own action execution, determines possible motor command sequences that bring about the last intermediate goal, the action goal. Although the primary goal differs between the human actor and the observer, the action goal does not. The action goal of the observer, and of the human actor, is that the object is held in a certain position. For the human actor, motor commands are required to achieve this action goal. The observer, however, executes an 'action' in the sense of a goal pursuit without motor commands. Since the other one is bringing about the action goal, there is no need for the observer to act himself. However, it may be relevant to monitor the goal pursuit, that is, the successful achievement of the intermediate goals. Under natural condition, this enables the observer to recognize a deviation from the expected sequence of intermediate goals, and to intervene in order to ensure the achievement of the action goal. Besides, even such a goal pursuit without own action allows the observer to have an expectation of the action goal value and to update this action goal value as the action goal is reached. This may be important for the later selection of own action goals. From the observer's point of view, the action goal of such an 'action' can be different from that of the actor depending on the subject's position in relation to the object. When the action goal is described in a world-centered frame of reference, it is the same.

According to this goal-pursuit-by-observer concept, as with the sensory concept, the predominance of non-shared codes and the rare occurrence of shared codes in our study can be explained by differences and similarities in the target position of the object during execution and observation from the monkeys' point of view. Depending on this target position, there are differences and similarities between the intermediate goals including the action goal. Shared codes would also be explained if intermediate goals were represented in a viewpoint-invariant manner. A number of previous findings can be explained by this concept. First, viewpoint-invariant encoding of an observed action has been shown in a subset of F5 MNs (*Caggiano et al., 2011*). Second, since the temporal sequence of intermediate goals is the same during execution and observation, the similarity of state transitions between execution and observation found by *Mazurek et al., 2018* can be explained if a state transition is interpreted as achievement of an intermediate goal and the activation of a subsequent intermediate goal. Notably, the finding of state transitions during action observation is similar to the distribution of the starting position of action segments in our study (see *Figure 5*): segments begin more often after the actor reaches the object and after the object is put to its target position, time points, which may correspond to state transitions. Third, that the monkey really pursues his own goal when observing is supported by the finding that the expected goal value determines how well he observes the action (*Pomper et al., 2020*). Fourth, the preparedness to act on one's own to achieve the goal in the case of the other's failure explains why some MNs are influenced by the reachability and spatial configuration of the observed action (*Caggiano et al., 2011*; *Caggiano et al., 2009*; *Maranesi et al., 2017*). Fifth, it is consistent with the representation of a goal pursuit during observation that information about the expected goal value for the observer is contained in F5 MNs (*Pomper et al., 2020*). Sixth, and most importantly, this information about the expected goal value explains the action goal selection deficit of monkeys with bilateral lesions of the subset of F5 neurons that project to the medial prefrontal cortex (*Ninomiya et al., 2020*). Monkeys with such lesions were less able to detect 'choice errors' of the observed actor. To detect a 'choice error', it is necessary to map the visual input of the observed action to an observer's own action goal and its expected value. Only if there is an expectation about the action goal value of an observed action can an error in action goal selection by the actor be detected and an incorrect updating of the action goal value, which influences a subsequent own action goal selection, be avoided. Thus, the study of Ninomyia et al. clearly shows that F5 neurons are neccessary for this mapping and/or the updating of the action goal value based on this mapping.

## Alternative concept 4: goal-pursuit-by-actor concept

The fourth alternative concept posits that the activity of F5 MNs during observation is a motor representation of the observed action, resorting to our broad definition of the mirror mechanism. The broad definition posits that, as with the sensory concept, the observed action is represented, but unlike the sensory concept, the representation contains information that we consider 'motor', that is, unique to action control. This information includes, but is not limited to, the action goal as an intended state, the action goal value, and the successful achievement of intermediate goals including the action goal. In this way, sensory input representing environmental changes is mapped onto a representation of a goal pursuit by the actor. This could allow the observer to perceive (understand) the process in the environment as goal pursuit and/or updating of own action goals or action goal values. In contrast to the goal-pursuit-by-observer concept, this goal-pursuit-by-actor concept assumes that the goal pursuit of the actor, but not that of the observer, is represented. We consider two possible distinctions from the strictly defined mirror mechanism. First, the observed action could be represented from the observer's point of view. In that case, the action representation in the observer would be predominantly different from that in the actor. Second, the observed action could, after all, be represented as if the observer were performing it in place of the actor, but not supramodally, as postulated by the strictly defined mirror mechanism, yet multimodally. That is, information of different sensory modalities is combined without reaching an abstract supramodal level. This account seems to be supported by the finding that the activity of some F5 MNs during execution depends on whether the action is performed in darkness or in light (*Maranesi et al., 2015*). However, this account does not explain the result of the Ninomyia et al. study that the value of an own action goal is updated by observing an action of the other, where the action goal from the observer's point of view is different from that from the actor's point of view. Therefore, it is more likely that the observed action is represented from the observer's point of view.

## Goal pursuit by actor or by observer?

Both concepts of a goal pursuit, that by the actor and that by the observer, explain our results and most of the findings discussed so far to a similar extent. They also have in common to emphasize a representation of action control components upstream of motor commands, for which there is good evidence (*Ferrari et al., 2005*; *Umiltà et al., 2008*). Hence, we currently do not consider either of them to be the better concept. But a critical experiment can be suggested for which the two concepts make different predictions. The experiment is similar to that of the Ninomyia et al. study (2020) with the difference that when observed, only the actor, not the observer, is rewarded differently, but in a way that is predictable to the observer. If the goal-pursuit-by-observer concept holds, the prediction is that F5 MNs of the observer represent only the expectation of the own reward and that a 'choice error' of the actor is not detected by the observer. However, if the goal-pursuit-by-actor concept holds, the prediction is that F5 MNs of the observer represent the expectation of the reward for the actor in the same way as the expectation of the own reward and that a 'choice error' of the actor is detected.

## The term 'mirror neuron'

Finally, the question must be addressed whether it is justified to hold on to the term 'mirror neuron'. The term suggests that these neurons are a motor representation of the observed action resulting from a mirror mechanism. As mentioned above, the term 'mirror mechanism' can be understood in different ways. Depending on the interpretation, the assessment of the evidence for whether MNs 'mirror' or not will differ. Moreover, several of the concepts discussed here, that offer an explanation for the activity of MNs during observation, contradict even a broader understanding of a mirror mechanism. An alternative, as some already practice, would be to speak of self-other-action-related neurons, which is a descriptive term that allows for different conceptual interpretations. However, if the focus is to be on neurons related to a mirror mechanism, then the mirror mechanism should be included in the definition of these neurons. For example, the cross-task classification could be used to define a 'mirror neuron', provided that the mirror mechanism is understood according to our strict definition.

## Conclusion

We conclude that our finding of a rare but more than random occurrence of time periods with shared action code does not rule out a motor representation of the observed action according to the strictly defined mirror mechanism in some F5 MNs at some time points. However, the predominance of non-shared codes with good population representation of observed actions renders alternative concepts more likely. Although it is certainly possible that several of the concepts discussed apply at different time points in different neurons, the current evidences suggest that many F5 MNs represent a goal pursuit from the observer's point of view, in which either the observer aims at the same action goal as the actor, or in which only the actor aims at a goal. Such a goal pursuit may allow the observer to use an expectation about the action goal value to optimize the updating of action goal values through observation alone, which is beneficial for subsequent own action goal selection. Which of the two possible variants of goal pursuits might be represented could be decided by the suggested future experiment.

# Materials and methods
## Experimental animals

The experiments were performed on two adult male rhesus monkeys (Macaca mulatta, age 8 and 9 years, respectively, median weight 10.6 kg and 8.4 kg, respectively). The animals lived in socially compatible groups with regular access to an open-air enclosure. Both animals had been obtained from the DPZ (Deutsches Primatenzentrum, Göttingen, Germany). All experiments were approved and controlled by the regional veterinary administration (Regierungspräsidium Tübingen and Landratsamt Tübingen, Permit Number: N4/14) and conducted in accordance with German and European law and the National Institutes of Health's Guide for the Care and Use of Laboratory Animals, and regularly and carefully monitored by the veterinary service of the University of Tübingen, the latter also providing care in case of medical problems.

## Behavioral tasks

Three objects to be manipulated by either the monkey or a human actor were fixed on a table and aligned parallel to the ground. The table rotated twice between trials (in the same or in opposite directions and for variable durations) to place randomly one of the three objects in front of the respective actor. The objects were metallic discs identical in shape and size (diameter 3.5 cm, height 0.5 cm). In the execution task, the center of the object at stake had a distance of 23 cm from the monkey's eyes. In the observation task, in which a human standing opposite to the monkey acted on the relevant object, the distance to the monkey was 49 cm. Each object accommodated only one specific type of action: it could be either lifted (2.7 cm up), twisted (50° clockwise), or shifted (1.9 cm to the right). Note that the identical appearance of the three objects denied any clue on the type of action they accommodated.

In the execution task, the rotation of the table and the execution of the action on the target object by the monkey took place in darkness. In the observation task, a light was switched on when the object was rotated in place so that the monkey could observe the experimenter standing opposite to him (light was turned off with object release or if a trial was aborted).

A trial started when the object was rotated in place and the actor (monkey in the execution task, human in the observation task) had pressed the start button (start button could be pressed before, during or after the rotation). A variable time (1–1.5 s) after trial start, an LED next to the positioned object that had halted in front of the respective actor, was turned on for 0.1–0.3 s. The LED color indicated the manipulation to be performed: green or yellow for a lift, white or blue for a twist, red for a shift. To evaluate whether the LED color was, indeed, used as a cue for action selection in the execution task, we randomly interleaved trials with a different LED, non-informative for the type of action, as a control in 5% of the trials: this LED was in a slightly different position and had a yellowish color that differed from the yellow of the informative cue. In these trials, the action was feasible by trying different manipulations on the object and using the resulting somatosensory feedback. The actor was allowed to release immediately the start-button to approach, manipulate, and finally hold the object against a resistance (against gravity for lift, against a spring for twist and shift). After a variable time after object hold start (0.3–0.8 s), the monkey — no matter if he had performed the action or had observed the human action — received water (or occasionally also units of a banana flavor high caloric drink) as reward, followed by the actor releasing the object. The amount of fluid per trial varied between 0.15 and 0.5 ml across sessions but was the same across the six actions (3 for each task) within a session.

The task events 'start-button press', 'start-button release', 'LED onset', 'touch of object', 'start moving the object', 'reaching the hold position', 'reward', and 'withdraw from the object' by releasing it, were registered by mechanoelectrical sensors. A trial was aborted (and a beep was given as auditory feedback to the monkey) if the start-button was released within 100 ms after LED onset (to motivate the monkey to use the LED cue), if the object was released before reward delivery or if a timeout was reached 10 s after trial start. To motivate the monkey even more to use the LED cue in the execution task, another timeout was active in 30% of trials (session-dependent rarely up to 100%) for the time period between 'touch of object' to 'start moving the object': 0.15 s (rarely 0.1 s) for a twist and a shift, 0.35 s (rarely 0.3 s) for a lift. The selection of time constraints and the proportion of trials in which they were applied, was a pragmatic compromise between a time limit, at which the LED signal had to be used as an informative cue for action selection in order to comply with the task, and a time span that allowed the task to be completed even when overall motivation was low. In the control trials with a non-informative LED signal no timeout was active. In the observation task, a trial was also aborted (and a beep was given as feedback to the monkey) if the monkey did not attend to the action as indicated by gaze not staying inside a given fixation window (see below). For experiment control and the recording of behavioral data, we deployed in-house open-source software (nrec: https://nrec.neurologie.uni-tuebingen.de, developed and maintained by F. Bunjes, J. Gukelberger, V. Gueler) running under Debian Linux on a standard PC.

## Measurement of eye movements

Eye position was recorded using an in-house video eye-tracker based on pupil detection in infrared light, operating at a sampling rate of 50 Hz, in one monkey and by permanently implanted search coils in the other monkey. Eye position recordings were calibrated at the beginning of each experimental

session by having the monkey fixate a dot displayed on a monitor, seen by the monkey in the table plane after redirection by a horizontally oriented mirror placed on top of the grasping table in front of the monkey. In the table plane, the target dot (red color, 0.1° radius) appeared within the range the human actor performed his action, allowing the reliable association of object position and eye movement records. As the monkey´s head was painlessly fixed (see below) during the experiment, eye position corresponded to gaze position. In the action observation task, attention to the action was ensured by making the delivery of reward dependent on gaze staying within a fixation window located horizontally at the center of the object, vertically 2.4° above the center of the front top edge of the object (from the monkey's viewpoint) and extending ± 7° vertically and ± 5° horizontally.

## Surgical procedures

The animals were implanted with a titanium post accommodating the painless fixation of the head and a titanium recording chamber overlying area F5 of the left hemisphere. The correct position of the chamber was determined using information from a pre-surgical anatomical MRI scan. All surgical procedures were conducted under strict aseptic conditions deploying combination anesthesia with isoflurane (0.8%) and remifentanil (1–2 microgram/kg·min) with full control of relevant physiological parameters such as body temperature, heart rate, blood pressure, $PO_2$, and $PCO_2$. Postoperatively, buprenorphine was given until signs of pain were gone. Animals were allowed to recover fully before starting the experiments.

## Electrophysiological recordings

Extracellular action potentials were recorded using glass-coated tungsten electrodes (0.5- to 2 MΩ impedance; Alpha Omega) using a multielectrode system equipped with up to eight probes (Alpha Omega Engineering). Action potentials of individual neurons were discriminated online resorting to template matching provided by Alpha Omega´s Multi Spike Detector.

The mirror neurons reported in this study were recorded from area F5 of the left hemispheres of the two experimental animals. Area F5 was targeted, guided by presurgical MRI, information on the location of the arcuate sulcus provided by electrode penetrations and a consideration of the response properties of neurons in F5 and in FEF as well as characteristic behavioral reactions to microstimulation with saccades evoked from the FEF and arm, hand, face, or mouth movements elicited when stimulating F5.

## Mirror neuron criteria

Unless mentioned otherwise, we used software based on MATLAB (2019a). For the analysis of discharge patterns, only well-isolated single units from area F5 were considered for which at least eight valid trials per condition were available. A trial was classified 'valid' if the aforementioned sequence of task events ('start-button press' to 'withdraw from the object') was registered by the sensors without technical malfunction. A neuron was classified as an F5 mirror neuron resorting to established criteria (e.g., *Pomper et al., 2020*). To this end, a task was subdivided into 5 epochs: baseline (–750 ms to –250 ms from LED onset), approach phase (from start-button release until touching the object), manipulation phase I (from touching the object until moving the object), manipulation phase II (from moving the object until holding the object in its target position), hold phase (from holding the object in its target position until 150ms later). A Friedman test with the factor 'epoch' was performed for each action (lift, twist or shift) separately. The neuron's response was classified as 'motor' or 'visual', if the Friedman test was significant for at least one action (Bonferroni correction, alpha = .05/3) in execution or observation, respectively. The classification as 'mirror neuron' was made, if a neuron had a motor and a visual response. Note that in accordance with standard procedures this classification did not require that the modulation affected the same epoch or same direction (that is discharge rate increase vs. decrease).

## Classification analyses

We used a linear discriminant classifier (*Fisher, 1936*) by employing the 'fitcdiscr' function (discriminant type: diaglinear) available in MATLAB 2021a, in order to explore if the information offered by either single mirror neurons or the population discharge, predicted specific actions (lift, twist, shift). Single neuron classifications were performed on a neuron's discharge rate in a given time bin. For the

population classification, these discharge rates were used as a feature of the classifier. The number of trials for each of the six conditions (two tasks times three actions) considered for single neuron classification was neuron-dependent and corresponded to the minimum number of trials per condition across both tasks and all the three actions. We performed an eight-fold cross validation method, which required that the minimum number of trials had to be adjusted to a multiple of eight. For population classification, the number of trials for each task condition was eight, since only neurons with at least eight trials per condition were considered (this also met the eight-fold cross-validation requirement). To obtain a confidence interval of mean accuracy for each classification, a bootstrapping procedure was applied (n=1,000 resamples). For each resample, as many trials were randomly selected with replacement from all trials of a condition as had been determined as the minimum number of trials before. The mean over eight accuracies obtained from eightfold cross validation was calculated and constitutes the accuracy assigned to a resample. For single neuron classification, we used the mean of the obtained distribution of accuracies as the neuron's classification accuracy in a certain time bin (*Figure 4* and *Figure 6A*). We conducted four types of classifications. The classifiers were trained on execution or observation trials and tested on the same task (within-task classification: e2e, o2o) or the other task (cross-task classification: e2o, o2e) respectively.

## Action encoding in single neurons

To test whether actions (lift vs. twist vs. shift) were encoded in a neuron at a certain time bin when observed and when executed, we deployed the Kruskal-Wallis test for each neuron and bin during execution (alpha <0.05, Benjamini-Hochberg-corrected for 177*24 tests) and during observation (alpha <0.05, Benjamini-Hochberg-corrected for 177*24 tests). For the following three methods to test for shared action code we considered time bins during the action period (from start-button release to reward). In each method, a test for shared action code was performed in single neurons for bins encoding actions during both execution and observation determined by the Kruskal-Wallis test described above.

## Shared action code in single neurons I, same discharge method

For each of the three actions a Wilcoxon rank sum test was applied bin-wise to test whether the discharge rate differs between the two tasks, execution and observation. The significance level of 5% was Benjamini-Hochberg-corrected for the number of tests (number of all bins in which actions were encoded times 3 for the three actions). If there was no significant difference for any of the three actions in a bin, the action-specific discharge rate was considered to be the same in both tasks.

## Shared action code in single neurons II, same preference method

For each bin, a positive action preference similarity index was determined according to the following four steps: (1) The mean discharge rate across trials was calculated for each of the three actions and for each of the two tasks resulting in two three-dimensional vectors, one for execution, one for observation. (2) Each vector was transformd so that the action with the highest discharge rate was given the value 1 and the other two were given the value 0. This was achieved by dividing by the highest discharge rate and then by rounding down. The case where the discharge rate was exactly the same or zero for all three actions was excluded by the preceding Kruskal-Wallis test, which had to be significant. (3) Each vector was normalized (by dividing by the length of the vector). (4) The dot product of the transformed and normalized vectors corresponds to the positive action preference similarity index. This index is 1 if the action preference is the same. It is 0 if different actions are preferred. The index is between 0 and 1 if two actions are preferred in one task but either only one of the two actions in the other task, or two actions in the other task with only one of the two being the same. To account for the case where one action had a lower discharge rate than the other two actions in both tasks, we also determined a negative action preference similarity index. This index was determined using the 4 steps described, with a different transformation in the second step. In order to give a value of 1 to the action with the lowest discharge rate, and a value of 0 to the others, the highest discharge rate was subtracted and the resulting vector was divided by its lowest value (with sign). Then, it was rounded down. We would like to point out that steps 3 and 4 of the determination of both indices are equal to the procedure described by *Mazurek et al., 2018* for the determination of a similarity index. Our index, however, is different, as it allows to unambiguously infer a same action preference

from an index value of 1 due to step 2. In addition, we excluded by the preceding statistical test that a value of 1 results from the unlikely but possible case that all discharge rates are exactly the same. To test for significance of same action preference we resorted to a bootstrapping procedure (n=1000 resamples). The bootstrap distribution and p-value were determined separately for the positive and negative action preference similarity index. The p-value was calculated by the relative number of resamples with an index below 1. The significance level of 5% was Benjamini-Hochberg-corrected for the number of tests (number of all bins in which actions were encoded times 2 for testing of positive and negative similarity index). If the positive or negative action preference similarity index of a bin had a significant value of 1 the same action preference was inferred.

### Shared action code in single neurons III, cross-task classification method using LDA

To test for significance of cross-task classification in a time bin, we used the bootstrap distribution of accuracy of the e2o classifier in this bin (see Classification analyses). A p-value was calculated by the relative number of resamples with an accuracy smaller or equal to chance level (1/3*100%). The significance level of 5% was Benjamini-Hochberg-corrected for the number of tests (number of all bins in which actions were encoded).

### Test against the null hypothesis that coding of actions in the population of MNs at observation is independent of the coding at execution

To test against the null hypothesis that the coding of actions in the population of MNs at observation is independent of the coding at execution, which we believe is consistent with the null hypothesis suggested by Gergely Csibra, who formulated it as follows: "... motor and visual properties are randomly distributed across neurons, with no systematic relationship between them." (*Csibra, 2005*), we performed the following permutation test (n=250 resamples). For each resample, each neuron's discharge rates of all trials of the three actions for execution were randomly assigned to a different neuron's discharge rates of all trials of the three actions for observation. It was ensured that there were not two resamples with the same remapping. The test for action encoding in single neurons and the three methods for shared action code were applied to the data of each resample to obtain method-specific null distributions for the number of neurons with shared action code in each time bin of the action period (from start-button release to reward).

### Comparison of accuracy in predicting observed actions between execution-trained and observation-trained classifiers

We compared the accuracies for predicting the observed actions of classifiers trained on execution (e2o classifiers) against that of classifiers trained on observation (o2o classifiers) for all bins of the action period (n=12*177=2,124 bins given n=177 MNs and n=12 bins of action period). Rather than using the absolute accuracies, we chose the difference relative to chance level, and plotted relative e2o accuracies vs. relative o2o accuracies (see *Figure 4*). In this way, data points indicating the same accuracies for both classifiers (e2o and o2o) fall on the unit line. Unequal accuracies lead to data points distributed around the zero line, which corresponds to the random level of e2o classifiers. In order to assess how data points are distributed with respect to these reference lines, we transformed each data point to polar coordinates and created a distribution of angles. Data points with an angle of 45 deg fall on the unity line and indicate a shared code, data points with 0 deg fall on the zero line. An obvious caveat is that meaningless data points are contained, characterized by the lack of action discrimination in the reference case, that is the o2o classification. As there is no a priori criterion that would allow one to distinguish good accuracy from insufficient accuracy, we compared the influence of increasingly restrictive, filtering thresholds on the o2o accuracy (starting from 5% above chance level to 15% in steps of 1%). For each threshold a one-Gaussian and a two-Gaussian fit was calculated for the histogram of the angles of all dots above this threshold using the function 'fitoptions' in MATLAB 2021a (method: non-linear least squares). The adjusted R-squared values and the residuals were calculated for both fits and are shown in *Figure 4*. For the two-Gaussian fit the peak values ('lower peak' and 'upper peak') and the trough value between them ('boundary') were determined. A 95%-confidence interval was calculated for adjusted R-squared, lower peak, upper peak and boundary by bootstrapping (original data, n=2124,, were resampled with replacement 1000 times).

Whenever the algorithm failed to find a fit with two distinct modes, the procedure was repeated with new initial guesses. This reduced the number of failed fits from 3.25% to 1.8%. The average boundary angle was calculated across the boundaries of the 11 thresholds.

## Acknowledgements

We acknowledge the contribution of Dan Arnstein in the design of the experiment and Dr. Peter Dicke for the presurgical MRT scans and implant design, assistance in surgeries and postsurgical care of animals as well as for his invaluable support in developing the experimental setup. We acknowledge support from the Open Access Publication Fund of the University of Tübingen.

## Additional information

### Funding

| Funder | Grant reference number | Author |
| --- | --- | --- |
| Deutsche Forschungsgemeinschaft | TH 425/12-2 | Jörn K Pomper Mohammad Shams Shengjun Wen Friedemann Bunjes Peter Thier |

The funders had no role in study design, data collection and interpretation, or the decision to submit the work for publication.

### Author contributions

Jörn K Pomper, Conceptualization, Data curation, Formal analysis, Supervision, Validation, Investigation, Visualization, Methodology, Writing – original draft, Project administration, Writing – review and editing; Mohammad Shams, Conceptualization, Data curation, Formal analysis, Validation, Visualization, Writing – original draft, Writing – review and editing; Shengjun Wen, Validation, Investigation, Writing – review and editing; Friedemann Bunjes, Software, Validation, Methodology, Writing – review and editing; Peter Thier, Conceptualization, Resources, Supervision, Funding acquisition, Validation, Project administration, Writing – review and editing

### Author ORCIDs

Jörn K Pomper  https://orcid.org/0000-0003-0926-0477
Mohammad Shams  https://orcid.org/0000-0002-5081-3427
Peter Thier  http://orcid.org/0000-0001-5909-4222

### Ethics

All experiments were approved and controlled by the regional veterinary administration (Regierungspräsidium Tübingen and Landratsamt Tübingen, Permit Number: N4/14) and conducted in accordance with German and European law and the National Institutes of Health's Guide for the Care and Use of Laboratory Animals, and regularly and carefully monitored by the veterinary service of the University of Tübingen, the latter also providing care in case of medical problems.

### Decision letter and Author response

Decision letter https://doi.org/10.7554/eLife.77513.sa1
Author response https://doi.org/10.7554/eLife.77513.sa2

## Additional files

### Supplementary files
• Transparent reporting form

### Data availability
The data and codes for reproduction of all figures and the table have been deposited on Zenodo at https://doi.org/10.5281/zenodo.8047592.

The following dataset was generated:

| Author(s) | Year | Dataset title | Dataset URL | Database and Identifier |
|---|---|---|---|---|
| Pomper JK, Shams M, Wen S, Bunjes F, Thier P | 2023 | Dataset Non-shared coding of observed and executed actions prevails in macaque ventral premotor mirror neurons | https://doi.org/10.5281/zenodo.8047592 | Zenodo, 10.5281/zenodo.8047592 |

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
