## [Editor Report]

The mechanisms underlying mirror neurons are a topic of wide interest for all who study the workings of the brain. The authors use an elegant and compelling decoding approach to test whether mirror neurons encode action categories in the same framework underlying action execution regardless of whether actions are executed in the dark or observed in the light. This new approach identifies only a very small subset of mirror neurons with fully matched coding among another set with partial matches and a population-wide code more consistent with representing goal pursuit. The thought-provoking and important study opens up new avenues to probe the neural mechanisms of matching action and perception.

---

## [Decision Letter]

**Decision letter after peer review:**

Thank you for submitting your article "Poor matching of action codes challenges ′mirroring′ in macaque F5 mirror neurons" for consideration by *eLife*. Your article has been reviewed by 3 peer reviewers, one of whom is a member of our Board of Reviewing Editors, and the evaluation has been overseen by Tirin Moore as the Senior Editor. The following individual involved in the review of your submission has agreed to reveal their identity: Luca Bonini (Reviewer #2).

Essential revisions:

All the reviewers were impressed by the experimental design and the elegant decoding method. They agreed that the results themselves are of broad interest to the neuroscience community. However, none of them felt that the results fully supported the conclusions drawn in the manuscript and all of them suggested reframing the results starting from the original motivation (title/abstract) forwards.

All the reviewers appreciate this is a wide-ranging recommendation. Please have a look at the reviewers' individual comments below as a guide to a potential reframing of the study.

Very briefly, one critical issue is that the reviewers did not think that the nature of the presented data could speak to the "understanding" of the actions of others and the results should be considered from a processing/representational perspective. Another issue is the requirement for a more differentiated consideration of the implications of the differences in cross-coding by different subgroups of mirror neurons. How does this relate to the existing literature and what does this mean for the potential functional operations of mirror neurons, with particular reference to the insights gained by using a decoding approach in this study.

*Reviewer #1 (Recommendations for the authors):*

1) Rethink the theoretical framework, especially with regard to how your results sit in relation to the more permissive, wider definition of "mirror neurons" used. I would suggest your present results directly challenge this.

2) Consider whether there is a way to check whether the decoding is directly relevant to the performed or observed actions. For instance, is coding performance better on trials with fast reaction times rather than trials where an animal perhaps first tries a different manipulation of the rigged object? Does decoder performance change depending on gaze position during object manipulation epoch (e.g. directly on hand/object rather than more peripherally)?

3) (lines 496-501) Can you give any more information about the functional representation / structural position of the F5 mirror neurons with the "full match"?

4) (lines 457-461) Discuss whether different time-outs for different actions could lead to neuronal activity differences that could aid the decoding of action categories.

*Reviewer #2 (Recommendations for the authors):*

1) Title. I strongly suggest changing the title with something equally provocative but conveying constructively the real achievement of the paper rather than fueling a sterile debate, e.g. "Premotor mirror neurons remap, rather than reflect, other's observed actions", or something similar, focused on what the data show. In my view, the concept of understanding must be abandoned for conceptual reasons.

2) The authors seem to overemphasize their theoretical claim by omitting to cite some of their own (e.g. Caggiano et al. 2009, Science) and others' (e.g. Maranesi et al. 2017; Mazurek and Schieber 2019 J Neurophysiol) works supporting the pragmatic role of mirror neuron activity. Furthermore, several recent experimental and review articles point to the same direction supported by the authors in this work (Ferroni et al. 2021 Curr Biol; Orban et al. 2021 TICS; Orban et al. 2021 Brain Struct and Funct), without even mentioning the term "understanding".

3) Page 4, line 41. "and OTHER species, including humans". Other is lacking, and the claim concerning "other species" should be supported by appropriate references.

4) Page 4, line 42. "Researcher have been mesmerized". This expression is useless and it seems to me to lack due respect for the work of the researchers that have made possible the present study with their discoveries.

5) Page 4, lines 47 to 51: these two sentences are meaningless, the recourse to the concept of "action understanding" is only a reuse of unjustified and tautological concepts erroneously introduced by others.

6) In line with the previous point, on page 4, lines 52 to 54, the authors now substitute the term "understanding" with that of "simulation" as if they were synonyms. It can certainly be the case, especially if, as I claimed, "understanding" is meaningless and can thus assume any possible meaning (please, compare this sentence with the sentence from lines 19 to 20 in the abstract).

7) Page 4, lines 54-55. "this is the tenet of the mirror mechanism (Rizzolatti and Sinigaglia, 2016)". This sentence (and many others along the paper) is only true if one (arbitrarily) decides to identify the mirror mechanism with what the cited paper claims, neglecting all the remaining literature that the authors do not actually consider, which points toward different directions.

8) Page 4, line 59. None of the studies listed here is about "neurons" at all, whereas several single-neuron studies in multiple areas, even authored by some coauthors of the present paper, provide direct evidence that the "so-called" mirror neurons and hence the mirror mechanism cannot be conceived as so rigid and strictly congruent as initially argued by the discoverers (see, e.g. Caggiano et al. 2016 Curr Biol; Papadourakis and Raos 2019; Lanzilotto et al. 2020 PNAS; Albertini et al. 2021 J Neurophysiol).

9) Page 5, lines 72-73. There is evidence that mirror neurons can also discharge when facing non-biological motion (Albertini et al. 2021 J Neurophysiol) when only the causal dynamic of an observed action is available (Caggiano et al. 2016 Curr Biol), or even in the complete absence of any motion at all (Bonini et al. 2014 Curr Biol): these issues could be considered.

10) Page 5, lines 76-79. Unless the authors think that there is an independent operational/behavioral definition of understanding, making justified the recourse to this concept, I don't think it can be claimed that any experimental data can shed any light on it. Certainly, their paradigm does not test "action understanding".

11) Page 5, lines 85-87. First, there is a circular claim in this series of (unjustified) assumptions (based on theoretical flaws): the sentence assumes that the paradigm conceived by the authors can substantiate the kind of representation of action goals supported by a mirror mechanism; second, it assumes that a mirror mechanism demands a matched code, whereas this is only implied by the original definition but not by several subsequent works. The question asked on lines 87-90 is the relevant one: is it possible to cross-decode observed actions using the neural code of executed actions? This has never been tested with different types of actions (but see Livi et al. 2019 PNAS), and it is a really novel and interesting aspect of this study.

12) When dealing with "action preference" (line 141-149), it is unclear if the author considered the 6 possible combinations for each modality mentioned in the Methods or only the three (as suggested by classification analysis – in this latter case it should be stated which was considered the "preferred" action among the two equally well encoded). Furthermore, in line 145 "the remaining neurons" is totally unclear as to what total is referring to. Thus, it remains unclear with this approach whether and to what extent the statement "only a minority of mirror neurons exhibited the same action preferences for execution and observation" is really justified.

13) Methods, page 17 line 442-444. Please clarify the meaning of the yellow cue: was it for the lift or was it noninformative for the type of object? And what the monkey did in front of it if yellow was used in – I guess – 50% of the lift trials? I can imagine the reason why different colors were used for the same object, but this should be explicitly stated in the methods.

14) Line 487. Remove "were monitored" (or rephrase the sentence).

15) The legend of Figure 4 to me is confusing. First, it talks about "a neuron in a time bin with respect to chance level", and immediately after, about the same panel A, it refers to "each neuron contributed to this plot with 12 time bins…". Please check and correct/clarify for consistency with the main text.

16) I would strongly suggest removing the statement in lines 340-343, and consider that the recently proposed concept of "social affordance" (Orban et al. 2021 TICS) seems to correspond to what the present data can nicely demonstrate.

*Reviewer #3 (Recommendations for the authors):*

The writing is mainly clear, the figures are clear, and I liked the many different analyses that addressed a variety of questions about subsets of neurons or subsets of time periods during the task, and so on.

My main problem with this study is the way it is presented as a test of a hypothesis, and as an argument against that hypothesis. I don't buy it. If this study were reported as simply data on mirror neurons, reporting what was found, with a proper discussion of the real caveats of the study, followed by the authors' opinion, recognizing that the study does not really support one or the other opinion, I would find it to be a useful minor replication of previous work on mirror neurons.

---

## [Author Response]

Essential revisions:All the reviewers were impressed by the experimental design and the elegant decoding method. They agreed that the results themselves are of broad interest to the neuroscience community. However, none of them felt that the results fully supported the conclusions drawn in the manuscript and all of them suggested reframing the results starting from the original motivation (title/abstract) forwards.All the reviewers appreciate this is a wide-ranging recommendation. Please have a look at the reviewers' individual comments below as a guide to a potential reframing of the study.Very briefly, one critical issue is that the reviewers did not think that the nature of the presented data could speak to the "understanding" of the actions of others and the results should be considered from a processing/representational perspective. Another issue is the requirement for a more differentiated consideration of the implications of the differences in cross-coding by different subgroups of mirror neurons. How does this relate to the existing literature and what does this mean for the potential functional operations of mirror neurons, with particular reference to the insights gained by using a decoding approach in this study.

Many thanks for the detailed and sometimes sharp, yet appropriate criticism of our study. It was an incentive for us to carry out additional analyses and to devote more effort to an elaboration of concepts. The outcome is that the results have changed slightly and that we now give more space to a discussion of concepts. We first address here the points raised by more than one reviewer before responding to comments contributed by individual reviewers.

The points raised can be divided into three thematic groups, (1) conceptual issues, (2) experimental and analytical questions, and (3) comments challenging the novelty of our results. On the first theme, we think it is essential to make a clear distinction between the conceptual and observational domains. As such, the criteria defining a “mirror neuron” and what is meant by the term "mirror mechanism" belong to the conceptual domain. This understanding of terms requires agreement among scientists, but is not experimentally testable. Unfortunately, there is no agreement on how to define a “mirror neuron” and what is meant by “mirror mechanism”. Thus, for the present work, the only option is to refer to specific definitions or to use our own, definitions which try to capture what others, and here most importantly Rizzolatti and colleagues, probably meant. We have adjusted the introduction in an attempt to convey our understanding and usage of the two terms in a hopefully comprehensible manner. Briefly, we use a definition for "mirror neuron" that we take from the first paragraph of the Results section of Gallese et al. (Brain, 1996). We do not consider the "properties of mirror neurons" described in that paper as defining a mirror neuron (MN). Classifying neurons as MNs only on the basis of the presence of a modulation of discharge rate during an executed and an observed action compared with a baseline is a common practice also in other single neuron studies on MNs, consistent with this definition. Regarding "mirror mechanism", we refer to Rizzolatti and Sinigaglia (2016) and make a distinction between a broad and a strict definition. Given our finding that there are almost no F5 MNs whose activity during observation is a motor representation according to our strict definition of a mirror mechanism, and also given the problem that the term “mirror mechanism” itself is not uniformly understood, the question arises whether and how the term "mirror neuron" should be used in the future. The answer to this may vary and belongs to the conceptual domain. We briefly address this question at the end of the discussion of the revised manuscript.

From that understanding of terms, conceptual hypotheses are to be distinguished, which of course must allow experimental predictions, i.e., must be falsifiable. We now distinguish more clearly between a "representation hypothesis" and an "understanding hypothesis". Both hypotheses focus on F5 MNs and are based on the strictly defined mirror mechanism. We test the “representation hypothesis” in our study, and just because it is the basis for the “understanding hypothesis”, falsifying the “representation hypothesis” would allow us to conclude that the “understanding hypothesis” is not valid. In contrast, confirmation of the “representation hypothesis” would not, of course, allow us to conclude that the “understanding hypothesis” holds. That would really be circular reasoning (this conclusion was drawn by some and rightly criticized). However, support for the “representation hypothesis” would be the necessary prerequisite for the “understanding hypothesis” to be true. These two hypotheses take up the original argument that a certain understanding of observed actions could follow from an equality of action-specific F5 MN activity during execution and observation. Because we considered the data on equality of action-specific F5 MN activity to be insufficient, we designed this study. Since our result largely argues against the "representation hypothesis" and thus against the "understanding hypothesis," we now discuss alternative concepts for the function of F5 MNs in more detail. It should be noted here that our fourth concept ("goal-pursuit-by-actor") could well represent the observed action without contradiction to our broad definition of a mirror mechanism, which in principle could also serve a subjective experience (which could be conceived as a kind of understanding). The way we structure the concepts in the discussion of this revised manuscript is, in our opinion, a useful overview of the concepts. The third concept is new in this context. We would like to emphasize that we focus on F5 MNs and intentionally avoid a discussion of mirror neurons beyond F5 in this paper. With the data from this study, we cannot say anything about MNs outside of F5.

Regarding the key question of how the "understanding hypothesis" is testable, or whether it may not be testable at all, we agree, of course, that for the conclusion of whether F5 MNs contribute to perception, only a manipulation of F5 MNs can clarify it. We now say that explicitly in the introduction. We agree with reviewer #2 that "understanding" here is not limited to "action recognition" or "action categorization”, which in principle could be implemented by purely sensory processing. Therefore, we also do not believe that the approach proposed by reviewer #3, which builds on the distinction of actions, would allow for a critical examination of the "understanding hypothesis”. But we disagree that the "understanding hypothesis" is not testable at all. Operationalization is necessary. If we accept that we can measure certain visual or auditory perceptions of an animal by operationalization (e.g., the subjective visual vertical, see for example Khazali et al., PNAS, 2020), then we must also accept that we can, in principle, measure other subjective experiences by operationalization, such as pain or aiming at a goal or even the co-experience of pain. An example of how to approach this is the study by Carrillo et al. (Curr Biol, 2019), which reviewer #2 and colleagues discussed in a recent review article (Bonini et al., TCS, 2022).

With regard to the second theme, experimental and analytical questions, we noticed while reading the comments that in our first version we did not distinguish clearly enough between statements about single neurons and statements about populations of neurons. Therefore, we now clearly separate single neuron analysis and population code analysis in the structure of the article. In view of the fact that statements about mirror neurons in the literature mostly refer to single neurons, we added extensive single neuron analyses, so that only now statistically reliable statements about single neurons are possible. This has led to the realization that the number of neurons with exclusively shared code is so small that these neurons should be considered a rare exception. Given the small number of time periods with shared code, we additionally tested against a hypothesis already rightly proposed as an alternative explanation by G. Csibra in 2005 (Mirror neurons and action observation: Is simulation involved? In: *What do mirror neurons mean?* Interdisciplines Web Forum 2005). We were able to reject this hypothesis based on two of three methods for testing for a shared code. This is the second piece of evidence besides the clustering of time periods with shared code already described in the first version that time periods with shared code cannot be considered random.

We discuss in more detail the question of whether neurons that exhibit a shared code at least at times support the representation hypothesis. To this end, we additionally examined whether certain action segments are more frequently represented with a shared than with a non-shared code, whether neurons with shared code differ from those with non-shared code in anatomical location, and whether an accuracy can be achieved with a time bin-wise selection of neurons with shared code by population cross-task classifiers as with within-task classifiers in the whole population.

Another issue was how to test for shared code and how to decide if a code has enough sharing. To answer the question, the exact hypothesis we intended to test here is crucial. The representation hypothesis states that the representation of the observed actions in F5 MNs corresponds to the representation as it occurs during the execution of the same actions. Therefore, the relationship between discharge rate and actions that holds during execution should also hold during observation, which is measurable with a classifier trained on execution trials and tested on observation trials. Moreover, the actions should not be more distinguishable during observation with a classifier other than the execution-trained classifier, because if that were so, it would mean that the representation of observed actions is different from that of executed actions. The detection of a cluster of time bins for which both conditions are satisfied confirms that it is possible to discover in this way the shared codes postulated by the representation hypothesis.

With respect to concerns that the monkey may not have used the cue at all when the action was executed, we added a comparison with control trials with a non-informative cue and also compared the duration of the approach phase between the three actions. Regarding oculomotor behavior, we verified that the monkey had actually directed his gaze toward the action during action observation for all three actions.

On the third issue, concerning the novelty of our results, we have now explained in more detail in the introduction why we felt it necessary to conduct a study we considered fundamental. As a result of our study, it can be clearly stated now that representations of observed actions as predicted by the strictly defined mirror mechanism are rare in F5 MNs, but nevertheless cannot be dismissed as random. This dispels the objection rightly raised by Csibra in 2005 and contradicts the currently prevailing view that such a representation can only be found at a population level. Even if these representations are ultimately explained by a concept other than the strictly defined mirror mechanism, their existence must be accounted for by any theory of the function of F5 neurons. Moreover, it is also shown that the observed actions are well discriminated with a non-shared code, at times even optimally. This contradicts the notion – which has been widespread for a long time since the work of Gallese et al. (Brain, 1996) – that mapping to motor representations in terms of broad congruence is simply not perfect. The applied cross-task decoding approach seems promising to test also in the future for a shared action code. Finally, reconsideration of alternative concepts has led us to highlight the possibility of a representation of a goal pursuit by the observer.

Reviewer #1 (Recommendations for the authors):1) Rethink the theoretical framework, especially with regard to how your results sit in relation to the more permissive, wider definition of "mirror neurons" used. I would suggest your present results directly challenge this.

Please see the response to the Essential revisions. We have integrated your suggestion.

2) Consider whether there is a way to check whether the decoding is directly relevant to the performed or observed actions. For instance, is coding performance better on trials with fast reaction times rather than trials where an animal perhaps first tries a different manipulation of the rigged object? Does decoder performance change depending on gaze position during object manipulation epoch (e.g. directly on hand/object rather than more peripherally)?

We agree that it is interesting to examine more closely the relationship between action discriminability and behavioral differences across trials. However, in our view, the proposed analyses in terms of reaction time go beyond the scope of this study and are only approximately able to answer the question of causality during action execution, i.e., whether neuronal activity controls behavior or is determined by changes in sensory input caused by behavior. In addition, because of the number of trials required for the analyses, only a subset of neurons could be used for the analyses.

Regarding the question of whether our result on shared codes could be influenced by different durations of the manipulation epoch, we would have to assume that the monkey did not use the cue in many trials, but tried different manipulations first. This is very unlikely, as the behavioral data show that the monkey did use the cue (Figure 1 D-F).

3) (lines 496-501) Can you give any more information about the functional representation / structural position of the F5 mirror neurons with the "full match"?

We plotted the anatomical locations of neurons with only shared and only non-shared codes (according to one of the methods) in Figure 7 and could not see any relevant differences.

4) (lines 457-461) Discuss whether different time-outs for different actions could lead to neuronal activity differences that could aid the decoding of action categories.

The difference in time-outs between the actions resulted from the fact that the "lift" action required more time. The time-outs were designed in such a way that they required the use of the cue (the actual use of the cue is now shown in Figure 1D-F). We therefore do not see how these different time-outs could have influenced our result on shared codes.

Reviewer #2 (Recommendations for the authors):1) Title. I strongly suggest changing the title with something equally provocative but conveying constructively the real achievement of the paper rather than fueling a sterile debate, e.g. "Premotor mirror neurons remap, rather than reflect, other's observed actions", or something similar, focused on what the data show. In my view, the concept of understanding must be abandoned for conceptual reasons.

We now limit ourselves in the title to the main finding.

2) The authors seem to overemphasize their theoretical claim by omitting to cite some of their own (e.g. Caggiano et al. 2009, Science) and others' (e.g. Maranesi et al. 2017; Mazurek and Schieber 2019 J Neurophysiol) works supporting the pragmatic role of mirror neuron activity. Furthermore, several recent experimental and review articles point to the same direction supported by the authors in this work (Ferroni et al. 2021 Curr Biol; Orban et al. 2021 TICS; Orban et al. 2021 Brain Struct and Funct), without even mentioning the term "understanding".3) Page 4, line 41. "and OTHER species, including humans". Other is lacking, and the claim concerning "other species" should be supported by appropriate references.4) Page 4, line 42. "Researcher have been mesmerized". This expression is useless and it seems to me to lack due respect for the work of the researchers that have made possible the present study with their discoveries.5) Page 4, lines 47 to 51: these two sentences are meaningless, the recourse to the concept of "action understanding" is only a reuse of unjustified and tautological concepts erroneously introduced by others.6) In line with the previous point, on page 4, lines 52 to 54, the authors now substitute the term "understanding" with that of "simulation" as if they were synonyms. It can certainly be the case, especially if, as I claimed, "understanding" is meaningless and can thus assume any possible meaning (please, compare this sentence with the sentence from lines 19 to 20 in the abstract).7) Page 4, lines 54-55. "this is the tenet of the mirror mechanism (Rizzolatti and Sinigaglia, 2016)". This sentence (and many others along the paper) is only true if one (arbitrarily) decides to identify the mirror mechanism with what the cited paper claims, neglecting all the remaining literature that the authors do not actually consider, which points toward different directions.

Points 2-7: The introduction has been thoroughly rewritten considering the suggestions and comments made by all three reviewers.

8) Page 4, line 59. None of the studies listed here is about "neurons" at all, whereas several single-neuron studies in multiple areas, even authored by some coauthors of the present paper, provide direct evidence that the "so-called" mirror neurons and hence the mirror mechanism cannot be conceived as so rigid and strictly congruent as initially argued by the discoverers (see, e.g. Caggiano et al. 2016 Curr Biol; Papadourakis and Raos 2019; Lanzilotto et al. 2020 PNAS; Albertini et al. 2021 J Neurophysiol).

As said, the introduction has been rewritten. All the studies mentioned are interesting. However, we cannot discuss all interesting studies within the scope of an article whose aim is the critical experimental and analytical examination of a properly defined question. And we also want to be very careful about not treating mirror neurons of different regions as one functional category.

9) Page 5, lines 72-73. There is evidence that mirror neurons can also discharge when facing non-biological motion (Albertini et al. 2021 J Neurophysiol) when only the causal dynamic of an observed action is available (Caggiano et al. 2016 Curr Biol), or even in the complete absence of any motion at all (Bonini et al. 2014 Curr Biol): these issues could be considered.

See point 8.

10) Page 5, lines 76-79. Unless the authors think that there is an independent operational/behavioral definition of understanding, making justified the recourse to this concept, I don't think it can be claimed that any experimental data can shed any light on it. Certainly, their paradigm does not test "action understanding".

As said, the introduction has been rewritten. And please see our response to the Essential revisions.

11) Page 5, lines 85-87. First, there is a circular claim in this series of (unjustified) assumptions (based on theoretical flaws): the sentence assumes that the paradigm conceived by the authors can substantiate the kind of representation of action goals supported by a mirror mechanism; second, it assumes that a mirror mechanism demands a matched code, whereas this is only implied by the original definition but not by several subsequent works. The question asked on lines 87-90 is the relevant one: is it possible to cross-decode observed actions using the neural code of executed actions? This has never been tested with different types of actions (but see Livi et al. 2019 PNAS), and it is a really novel and interesting aspect of this study.

See point 8.

12) When dealing with "action preference" (line 141-149), it is unclear if the author considered the 6 possible combinations for each modality mentioned in the Methods or only the three (as suggested by classification analysis – in this latter case it should be stated which was considered the "preferred" action among the two equally well encoded). Furthermore, in line 145 "the remaining neurons" is totally unclear as to what total is referring to. Thus, it remains unclear with this approach whether and to what extent the statement "only a minority of mirror neurons exhibited the same action preferences for execution and observation" is really justified.

This comment as well as the previous question about rescaling, the demand for better statistics and also the need to better compare with the studies of Mazurek et al. (J Neurosci, 2018) and Papadourakis and Raos (Cereb Cortex, 2019), led us to conduct an extended, statistically sound analysis on "same preference", which we present in parallel to the LDA single neuron analysis in the article now.

13) Methods, page 17 line 442-444. Please clarify the meaning of the yellow cue: was it for the lift or was it noninformative for the type of object? And what the monkey did in front of it if yellow was used in – I guess – 50% of the lift trials? I can imagine the reason why different colors were used for the same object, but this should be explicitly stated in the methods.

Our previous description was not quite correct. The color of the non-informative cue was a different yellow than the yellow of the informative cue. The non-informative cue was also in a different position. We adapted the text.

14) Line 487. Remove "were monitored" (or rephrase the sentence).

Done. Thanks.

15) The legend of Figure 4 to me is confusing. First, it talks about "a neuron in a time bin with respect to chance level", and immediately after, about the same panel A, it refers to "each neuron contributed to this plot with 12 time bins…". Please check and correct/clarify for consistency with the main text.

Our description was correct in content, but obviously not easy to understand. We have rearranged the sentence a bit and hope that it is easier to understand this way. In any case, it should be clear from the main text what is meant.

16) I would strongly suggest removing the statement in lines 340-343, and consider that the recently proposed concept of "social affordance" (Orban et al. 2021 TICS) seems to correspond to what the present data can nicely demonstrate.

The discussion has been rewritten. The paper was considered.

Reviewer #3 (Recommendations for the authors):The writing is mainly clear, the figures are clear, and I liked the many different analyses that addressed a variety of questions about subsets of neurons or subsets of time periods during the task, and so on.My main problem with this study is the way it is presented as a test of a hypothesis, and as an argument against that hypothesis. I don't buy it. If this study were reported as simply data on mirror neurons, reporting what was found, with a proper discussion of the real caveats of the study, followed by the authors' opinion, recognizing that the study does not really support one or the other opinion, I would find it to be a useful minor replication of previous work on mirror neurons.

Please see our response to the Essential revisions. We have also addressed your criticism by giving much more space to the alternative concepts and also clarifying that our study alone does not substantiate any of the four alternative concepts. However, we believe that our study not only provides new results, but also helps to remove conceptual ambiguities.